# Brain-inspired $L_p$-Convolution benefits large kernels and aligns better with visual cortex

## Abstract

Convolutional Neural Networks (CNNs) have profoundly influenced the field of computer vision, drawing significant inspiration from the visual processing mechanisms inherent in the brain. Despite sharing fundamental structural and representational similarities with the biological visual system, differences in local connectivity patterns within CNNs open up an interesting area to explore. In this work, we explore whether integrating biologically observed receptive fields (RFs) can enhance model performance and foster alignment with brain representations. We introduce a novel methodology, termed $L_p$-convolution, which employs the multivariate $p$-generalized normal distribution as an adaptable $L_p$-masks, to reconcile disparities between artificial and biological RFs. $L_p$-masks finds the optimal RFs through task-dependent adaptation of conformation such as distortion, scale, and rotation. This allows $L_p$-convolution to excel in tasks that require flexible RF shapes, including not only square-shaped regular RFs but also horizontal and vertical ones. Furthermore, we demonstrate that $L_p$-convolution with biological RFs significantly enhances the performance of large kernel CNNs possibly by introducing structured sparsity inspired by $p$-generalized normal distribution in convolution. Lastly, we present that neural representations of CNNs align more closely with the visual cortex when $L_p$-convolution is close to biological RFs. This research shines a light on the potential of brain-inspired models that merge insights from neuroscience and machine learning, with the hope of bridging the gap between artificial and biological visual systems.

## 1 Introduction

Convolutional neural networks (CNNs) were initially inspired by the early discoveries from the brain's visual system, particularly the groundbreaking work of Hubel and Wiesel on receptive fields (RFs) in the primary visual cortex (V1) (Hubel & Wiesel, 1962; 1965). This laid the foundation for the Neocognitron, a precursor to CNNs, which focused on the idea of local connections (Fukushima, 1980). Building upon the backpropagation (Werbos, 1974; Rumelhart et al., 1986), and the concept of weight sharing, LeCun and his colleagues applied it to CNNs, leading to the development of the first practical CNN, known as LeNet (LeCun et al., 1989). Fueled by advancements in backpropagation, large datasets, and GPU training, the landmark success of AlexNet in 2012, which significantly outperformed existing models in the ImageNet challenge, marked a turning point for CNNs (Krizhevsky et al., 2012).

In recent times, CNNs have become a prominent model for understanding biological visual processing due to both structural and representational similarity between them (Lindsay, 2021). The CNN's architecture closely mirrors the visual system: 1) RGB channels emulate retinal computations, 2) stacked layers correspond to various visual areas (such as V1, V2, V4, IT), 3) deeper layers excel at detecting complex features, and 4) local receptive fields. In terms of representational similarity, Khaligh-Razavi & Kriegeskorte (2014) demonstrated that among 37 different computational models, CNN trained with supervision exhibited the best performance and alignment with the brain. Recently, CNNs like VGG and ResNet have become instrumental models for visual processing models in neuroscience, owing to their strong representational alignment with the brain (Khaligh-Razavi & Kriegeskorte, 2014; Simonyan & Zisserman, 2014; Shi et al., 2019; He et al., 2016; Bakhtiari et al., 2021).

While CNNs employed the concept of local connectivity inspired by the brain, they are not exact replicas. This is attributed to the contrast between the dense, uniformly connected local RFs in CNNs and the varied and sparsely connected patterns in biological neurons. This disparity poses an intriguing question that could mutually captivate both neuroscience and machine learning communities: **Would integrating biologically observed receptive fields into an artificial model result in enhanced performance and closer alignment with brain representations?**

In this paper, we introduce $L_p$-convolution, a novel approach that leverages the multivariate $p$-generalized normal distribution (MPND) to address the disparities between biological and artificial RFs (Fig. 1). Through channel-wise trainable $L_p$-masks in convolutional layers (Fig. 2), we explore their conformational adaptability (Fig. 3 and 4), resulting in enhanced performance in large kernel CNNs and improved alignment with biological representations (Fig. 5 and Table 1).

Overall, our contributions are:

- Introduction of MPND to model the disparity between biological and artificial RFs.
- Proposal and implementation of $L_p$-convolution leveraging adaptable $L_p$-masks.
- Demonstration of task-dependent conformational adaptability of $L_p$-masks with Sudoku challenge.
- Significant performance gains in large kernel CNNs in vision classification tasks.
- Improved alignment of neural representations with brain-inspired $L_p$-convolution.

Code, datasets, and pre-trained models are available at `https://anonymous.4open.science/r/lpconv-E39D`.

## 2 BRIDGING BIOLOGICAL AND ARTIFICIAL RECEPTIVE FIELDS

The distinct characteristics of dense, uniform local RFs in CNNs, as opposed to the sparse, morphologically structured connectivity patterns in biological neurons, pose a challenge for directly comparing their local RFs. To address this, we analyzed the functional connectivity patterns of both biological and artificial systems, by introducing the MPND (Goodman & Kotz, 1973). MPND is the key of our paper to bridge the conformational difference between biological and artificial RFs. While the concept of the receptive field generally encompasses sensory-level inputs and adjacent layers, in this paper, we specifically refer to **RF as the local connectivity patterns** between neurons in adjacent layers.

**Multivariate $p$-generalized normal distribution**   Let $\mathbf{s}$ represent the $d$-dimensional random vectors indicating specific points within RF. $s_0$ is the receptive center with $d$-dimensional vector of fixed constants. The relative offset is given by $\Delta\mathbf{s} = \mathbf{s} - s_0$. Introducing MPND, we define the RF using as probability density function (PDF) of $\mathbf{s}$ as following:

$$\beta \exp\left(-\|\mathbf{C}\Delta\mathbf{s}\|_p^p\right), \tag{1}$$

where $\mathbf{C}$ is $d \times d$ covariance matrix, $\|\cdot\|_p^p$ denotes the $L_p$-norm raised to the $p$-th power, $\beta$ is normalization factor[1], and $p \geq 1$. In Figure 1e, we show some examples of MPNDs with varying values of $p$ and $\mathbf{C}$.

**Constructing biological and artificial RFs from the functional synapses**   To compare the biological and artificial RFs, we first prepared the 2D offsets of functional synapses relative to the post-synaptic units in both systems: mouse V1 L2/3 and AlexNet Conv1 (Fig. 1a-d; See details in Appendix A.1 and A.2). To match the scale difference between the two systems, we standardized the relative offsets with zero mean and unit variance. For the artificial system, we prepared 4 different cases, using both ImageNet-1k pre-trained and randomly-initialized AlexNet[2] with inputs of noises or images (See inputs and corresponding RFs in Appendix A.4). We constructed 2D probability mass functions (PMFs) from the collected offsets, which we call biological or artificial RFs.

---

[1] $\beta = [(2\Gamma(1 + p^{-1}))^d \cdot \det(\mathbf{C})]^{-1}$ where $\Gamma$ is gamma function, and $\det(\cdot)$ denotes the determinant.

[2] For clarity, we refer to *trained* as pre-trained models and to *untrained* as randomly-initialized models.

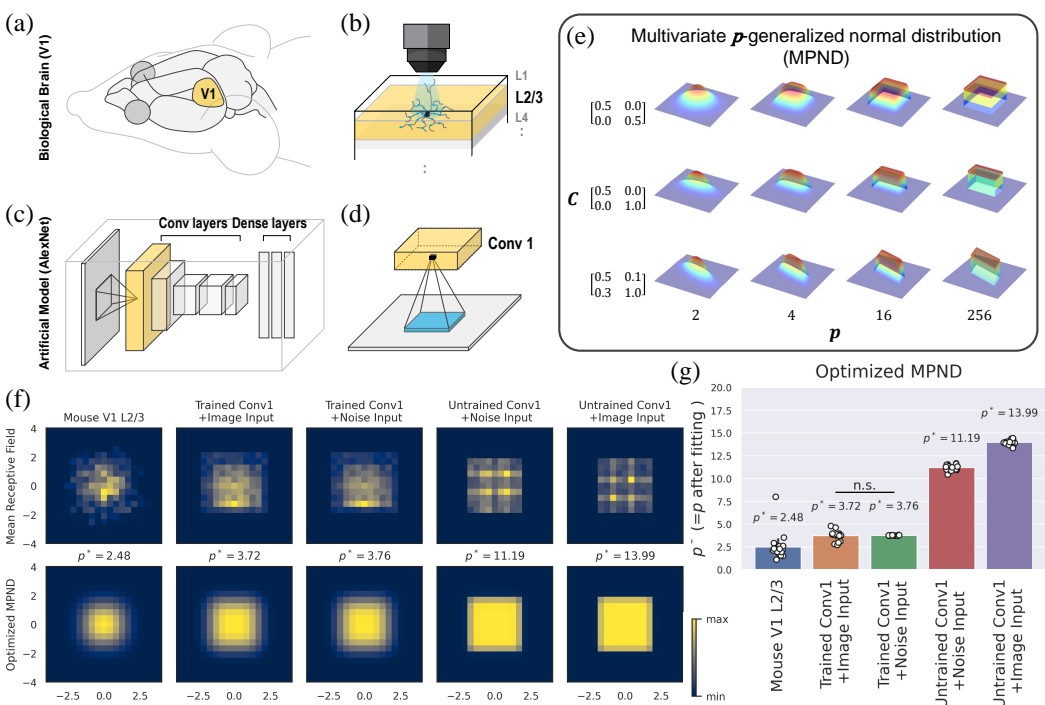

Figure 1: **Local receptive fields from biological and artificial systems can be mathematically reconciled by introducing multivariate $p$-generalized normal distribution** (a-b) Graphical illustration of receptive fields in V1 of mouse brain (a) at layer 2/3 (b). (c-d) Graphical illustration of receptive fields in AlexNet (c) at Conv1 layer (d). (e) Shapes of MPND with varying parameters of $\mathbf{C}$ and $p$. (f) Top, visualization of mean receptive fields of functional synapses in mouse V1 layer 2/3 (column 1) and AlexNet Conv1 with varying conditions (columns 2-5). Bottom, optimized MPND over receptive fields shown in the first row. (g) $p$ after MPND optimization; Using Welch's t-test with Holm-Bonferroni's multiple comparisons correction, all possible combinations between groups were statistically significant (p-value<0.05) except for 'n.s.' (non-significant) denoted in the figure; n=17 for all conditions. We optimized MPND parameters of $p$ and $\sigma$, where $\mathbf{C} = \begin{bmatrix} 1/\sigma & 0 \\ 0 & 1/\sigma \end{bmatrix}$, $\sigma_{\text{init}} = 0.5$, and $p_{\text{init}} = 2$

**MPND effectively models both biological and artificial RFs** For the comparison of biological and artificial RFs, we optimized parameters of $p$ and $\sigma$ in MPND (Fig. 1e, Eq. 1) over PMFs of biological or artificial RFs (Fig. 1f and g). We show that optimized $p^*$ of functional synaptic input patterns of biological neurons were optimized at near 2 (Gaussian-distributed; See details in Appendix A.3). In contrast, the local RF pattern of pre-trained AlexNet's Conv1 was optimized at the range of $3.7 \sim 3.8$, and the untrained one was optimized at the range of $11 \sim 14$ (where input types were less effective). An intriguing observation is the decrease in the value of $p$ for the pre-trained AlexNet's Conv1, bringing it closer to the biological RF. Based on these findings, we propose that both biological and artificial RFs can be effectively modeled with MPND, particularly within the range of $p = [2, 16]$. Given these findings, we propose to consider that the value of $p$ close to 2 is indicative of biological RF, while a higher $p$ represents RFs to be more artificial.

## 3 $L_p$-CONVOLUTION: INTRODUCING MPND IN THE CONVOLUTION

Based on our observation in Figure 1 artificial RFs become closer to biological RFs with training (decrease in $p$ of MPND by ImageNet-1k training in AlexNet Conv1), we asked two intriguing questions: 1) Is it possible to improve the performance of CNNs by implementing biologically-inspired RFs on artificial models? 2) Can CNNs with RFs close to biological ones align better

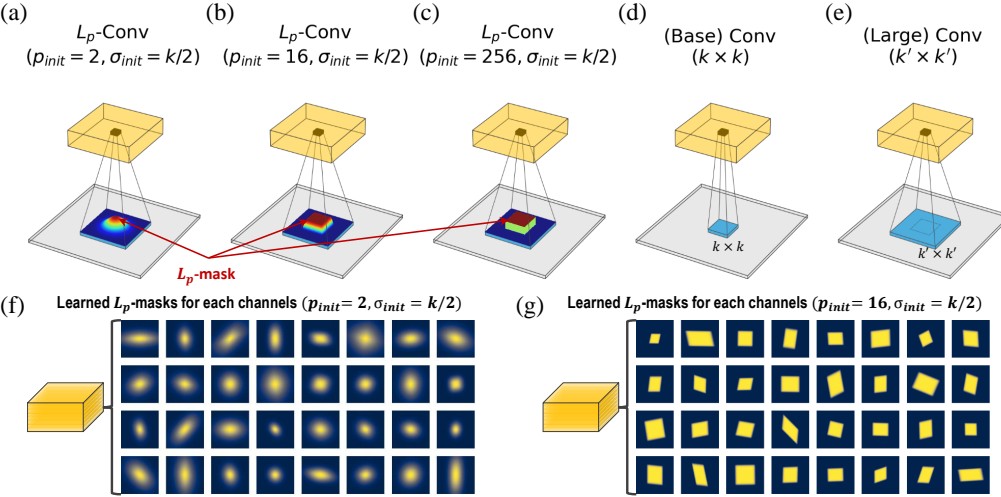

Figure 2: **Graphical illustration of $L_p$-Conv layers and visualization of learned $L_p$-masks after traning** (a-c) $L_p$-Conv layers with $\sigma_{\text{init}} = k/2$ and varying $p_{\text{init}} = \{2, 16, 256\}$; $L_p$-masks overlaid on kernels (red arrows). (d-e) Conventional Conv layers with the kernel size of $k \times k$ (Base) or $k' \times k'$ (Large). (f-g) Visualized 32 example learned $L_p$-masks from $L_p$-converted AlexNet Conv1 after training with Tiny Imagenet dataset

with the representation of the brain? To answer these questions, we introduce $L_p$-**convolution**: overlaying channel-wise trainable $L_p$-masks onto the kernels of CNNs.

$L_p$-**convolution**    Here, we propose the $L_p$-convolution, which is compatible with various convolutions. We formulate the $L_p$-convolution based on the MPND in the convolutional layer by employing channel-wise $L_p$-masks, which are overlaid on convolutional filters (Fig. 2a-c).

First, we define the relative height and width offsets, $\Delta \mathcal{S} \in \mathbb{R}^{2 \times K_h \times K_w}$ from the kernel center, $(K_h/2, K_w/2)$, as follows

$$\Delta \mathcal{S}_{\cdot,h,w} = (\Delta h, \Delta w)^T = (h - \frac{K_h}{2}, w - \frac{K_w}{2})^T \text{ for } h \in [0, K_h - 1], w \in [0, K_w - 1], \quad (2)$$

where $K_h$ and $K_w$ denote the kernel height and kernel width, respectively. Empirically, we utilize the normalized value between 0 and 1. It is noted that $\mathcal{S}_{\cdot,h,w} \in \mathbb{R}^2$ denotes all values corresponding to $h$-th height and $w$-th width.

Second, we propose the $L_p$-mask, structured mask matrix, derived from the offset and MPND. Our $L_p$-mask, $\mathcal{M} \in [0, 1]^{C_o \times K_h \times K_w}$ for all output channel $C_o$, is a soft mask that is proportional to Eq. 1 without a normalization factor $\beta$ as following

$$\mathcal{M}_{o,h,w} = \exp\left(-\|\mathcal{C}_{o,\cdot,\cdot} \Delta \mathcal{S}_{\cdot,h,w}\|_p^p\right), \quad (3)$$

where $\mathcal{C} \in \mathbb{R}^{C_o \times 2 \times 2}$ is the set of $2 \times 2$ covariance matrix for each output channel. In other words, $L_p$-mask calculates the soft mask $\mathcal{M}_{o,\cdot,\cdot}$ for each $o$-th output channel independently, and each soft mask handles the positional correlation, height and width position, from the offset, $\Delta \mathcal{S}$. Our $L_p$-mask in Eq. 3 corresponds to RF in Eq. 1, when $K_h = K_W = C_o = 1$.

Third, we propose the $L_p$-convolution by applying $L_p$-mask into the convolutional weights $\mathcal{W} \in R^{C_i \times C_o \times K_h \times K_w}$, where $C_i$ is the number of input channel. For each $i$-th input channel input $\mathcal{X}_i$ and convolutional filter weights $\mathcal{W}_i$, we formulate the corresponding convolution output $\mathcal{Y}_i$ as

$$\mathcal{Y}_i = \phi(\mathcal{X}_i * (\mathcal{W}_i \odot \mathcal{M})), \quad (4)$$

where $\phi$ is non-linear function and $*$ is the convolution operation.

We note that $\mathcal{C}$ and $p$ are trainable parameters[3]. To ensure the positive definite property of $\mathcal{C}_o$ and satisfy the $L_p$-norm property with $p \geq 1$, we can employ Cholesky decomposition for $\mathcal{C}$ and value clipping for $p$.

As shown in Equation 4, our $L_p$-convolution is a generalized version of the traditional convolution. In our settings, it is noted that $L_p$ mask converges to a binary mask as $p$ approaches infinity. Empirically, $L_p$ mask becomes a binary mask for sufficiently large $p$, as shown in Fig. 1. If all elements of $L_p$ mask equal one, then $L_p$-convolution degrades to the traditional convolution. In this situation, both traditional convolution and our $L_p$ convolution have square-shaped RFs for each layer. In other words, our $L_p$-convolution takes task or data-dependent RFs with varying $\mathcal{M}$ by optimizing the $\mathcal{C}$ and $p$ as shown in Figure 2. Therefore, $L_p$-convolution has task-specific RFs with varying distortion, scale, and rotation levels.

In practical terms, we replaced all existing Conv2d layers in the baseline CNN model with $L_p$-Conv layers by applying a function called **LpConvert** to the baseline CNN model (See pseudo-code in Appendix A.13). To provide further insight into the conformational changes of $L_p$-masks during model training, we present examples of 32 random $L_p$-masks from Conv1 of an AlexNet model trained with TinyImageNet (f and g in Fig. 2).

## 4    CONFORMATIONAL ADAPTABILITY OF $L_p$-MASKS IN SUDOKU CHALLENGE

To assess the adaptability and effectiveness of evolving RFs in $L_p$-convolution, we conducted well-established $9 \times 9$ Sudoku-solving task (Park, 2018; Oinar, 2021; Amos & Kolter, 2017; Palm et al., 2018; Wang et al., 2019) (See experimental details in Appendix A.5). We selected the Sudoku task for its challenging nature, as it necessitates simultaneously achieving three objectives—ensuring every row, column, and box contains all numbers from 1 to 9—(Fig. 3a).

$L_p$**-convolution in Sudoku solving: balancing square and row-column imbalances**    As shown in Figure 3b, the ($3\times3$) Base model, or ($7\times7$) $L_p^{\dagger}(p_{\text{init}} = 256)$ model exhibited a significant imbalance in Square-to-Row/Column accuracy and showed signs of overfitting after approximately 15 epochs. The ($7\times7$) Large model demonstrated improvement in the balance of Square-to-Row/Column accuracy and overall Sudoku accuracy, possibly due to the enlarged RFs. Along with these control experiments, we tested two trainable $L_p$-masks with $p_{\text{init}} = 2, 16$, each of them closely resembles a biological RF ($p = 2$) and an artificial RF in ($p = 16$), respectively. In ($7\times7$) $L_p(p_{\text{init}} = 2)$ model, we observed highly balanced Square-to-Row/Column accuracy, resulting in remarkable improvement in overall Sudoku accuracy compared to ($7\times7$) $L_p(p_{\text{init}} = 16)$ or ($7\times7$) Large model (red and green in Fig. 3b). We speculated that this alleviation of Square-to-Row/Column accuracy imbalance in ($7\times7$) $L_p(p_{\text{init}} = 2)$ could be attributed to the task-dependent $L_p$-masks' conformational adaptability. To test this possibility, we have designed ablation experiments on ($7\times7$) $L_p(p_{\text{init}} = 2)$ model.

**Ablation of orientation selective masks reveals $L_p$-masks' conformational adaptability**    Contrary to a previous large-kernel model that introduces unstructured sparsity directly into filters (Liu et al., 2023), $L_p$-convolution with $p = 2$ introduces structured sparsity based on a Gaussian distribution. This approach facilitates covariance analysis of the Gaussian distribution, thereby enhancing interpretability. Using Singular Value Decomposition (SVD) on $\mathcal{C}$, we extracted three interpretable properties of scale ($\alpha$), rotation ($\theta$), and distortion ($\gamma$) (See conformational analysis in Appendix A.5). Figure 4a shows conformations of $L_p$-masks inverse calculated from $\alpha$, $\theta$, and $\gamma$.

Quantitative analysis of ($7\times7$) $L_p(p_{\text{init}} = 2)$ model revealed an increase in scales when layer deepened (Fig. 4b, see visualization in Appendix A.7), with orientations of horizontal (0,180°) and vertical (90°) directions. This indicates the task-dependent adaptation of $L_p$-masks, which provide flexible and structured RFs in visual processing. To confirm these orientation-selective masks contribute to the balanced Sudoku-solving task, we conducted an ablation test. We classified masks with highly distorted ($\gamma > 3$) as orientation-selective masks. Among these, we selectively ablated near 90° by gradually increasing the range (close in shape with red dashed box in Fig. 4a) while

---

[3]$\mathcal{C}$ and $p$ are updated with the standard backpropagation process. $L_p$-mask, $\mathcal{M}$, is dynamically generated during forward process using $\mathcal{C}$ and $p$.

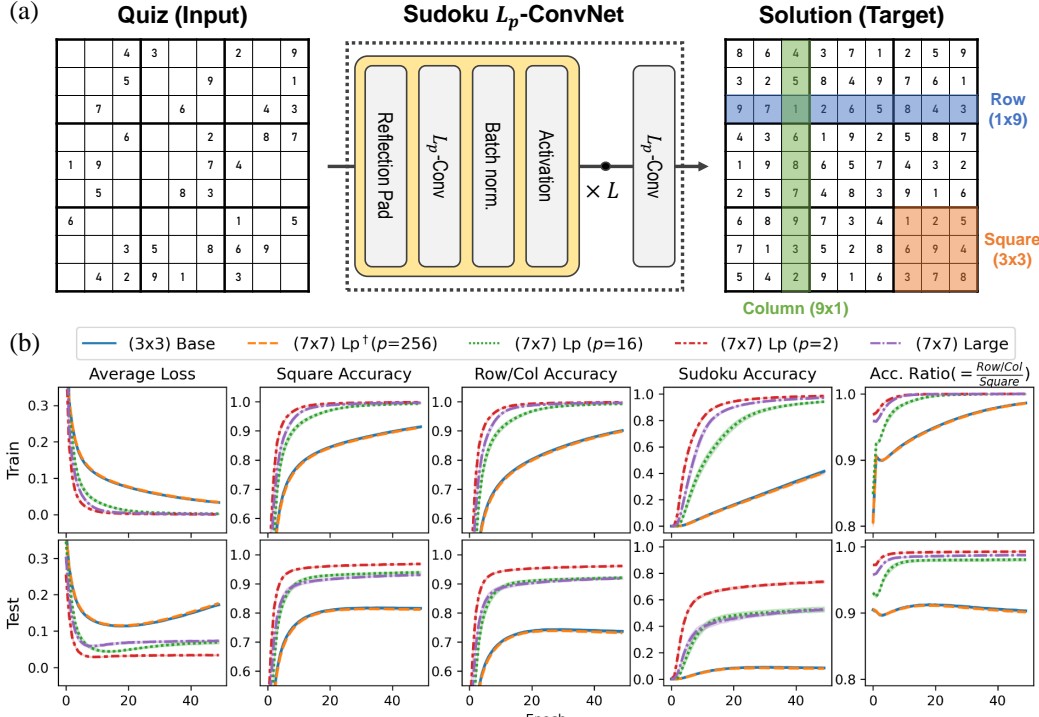

Figure 3: $L_p$**-convolution enhances Sudoku solving efficiency by effectively balancing accuracy between square and row-column puzzles** (a) Graphical illustration of Sudoku quiz and Sudoku $L_p$-ConvNet; (left), example Sudoku quiz as an input; (middle), basic block repeated $L$ times ($L = 10$, yellow) contains sequential layers of 1) reflection padding, 2) $L_p$-Conv, 3) batch normalization, and 4) activation layers; (right), the example Sudoku solution as a target. In Sudoku, a 9×9 square must be filled in with numbers from 1-9 with no repeated numbers in 9x1 rows (blue), 1x9 columns (green), or 3x3 squares (orange). (b) Loss and accuracy curves during training and test sessions. '$(3 \times 3)$' or '$(7 \times 7)$' denotes the size kernel. '$L_p^{\dagger}$' denotes parameters of $L_p$-mask is frozen. 'Large' denotes a simple enlargement of the kernel, without a mask.

tracking changes in column and row accuracies. While row and box accuracy exhibited a consistent decrease, column accuracy sharply decreased as the $\theta$ range increased (Fig. 4c and Appendix A.8), with this trend was notable in the later layers. Together, these results indicate that the conformational adaptability of $L_p$-masks enables balanced learning in the Sudoku-solving task, thereby contributing to overall performance enhancement.

## 5  $L_p$-CONVOLUTION BENEFITS LARGE KERNEL CNNS IN VISION CLASSIFICATION TASK

Next, we asked whether our designed $L_p$-convolution would be beneficial for enhancing the performance of historically successful CNNs. To test this, we conducted vision classification tasks (See detailed experimental settings in Appendix A.9) using the CIFAR-100 and TinyImageNet datasets on models of AlexNet, VGG-16, ResNet-18, ResNet-34, and the ConvNext-tiny (Liu et al., 2022) (Table 1). When simply increasing the kernel size, we observed a significant decrease in performance for all models except ResNet. This is consistent with a previous report of a decrease in vision classification performance with larger kernels (Peng et al., 2017).

As depicted in Table 1, when parameters are frozen with a notably large $p$ ($p = 256$ and $\sigma = \frac{k}{2}$), $L_p$-Conv exhibits performance comparable to the baseline model (refer to subfigures c and d in Fig. 2 for graphical illustration). This suggests that initializing the parameters of $L_p$-Conv with a similar experimental configuration can contribute to the stable learning of $L_p$-masks, avoiding significant

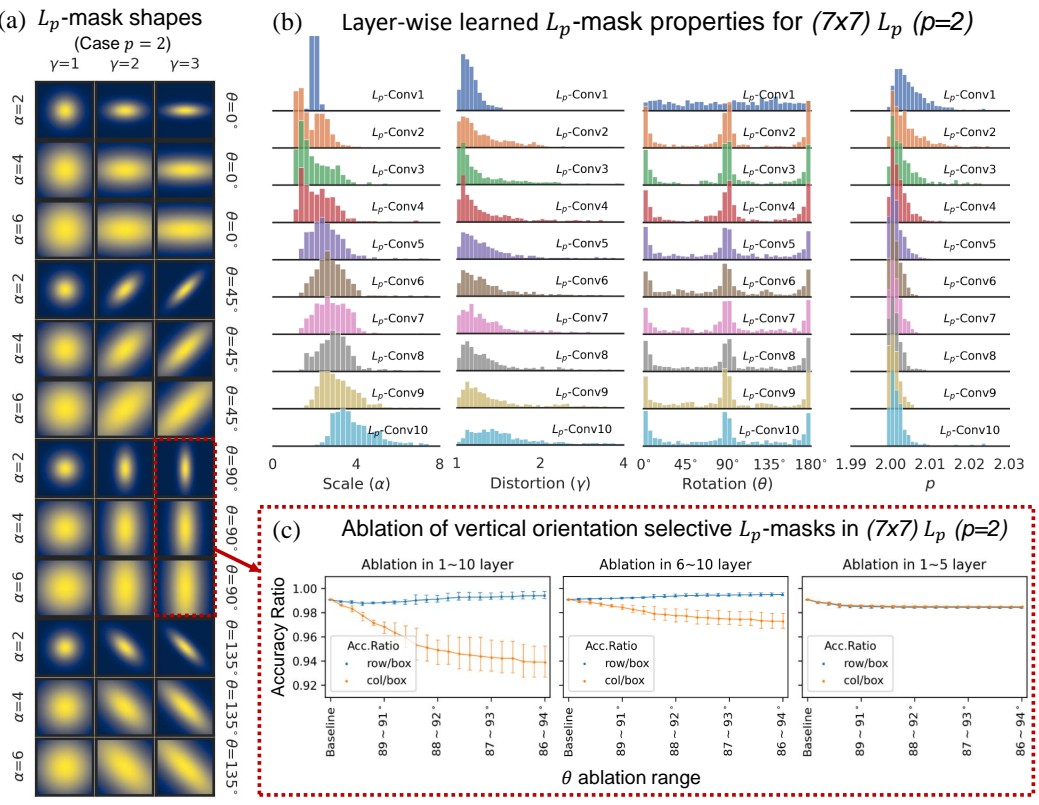

Figure 4: **Task-dependent conformational adaptation of $L_p$-masks** (a) The shapes of $L_p$-masks when $p = 2$ and varying properties of scale ($\alpha$), distortion ($\gamma$), and rotation ($\theta$), which are derived from the singular value decomposition of $\mathcal{C}$; Red box indicates column selective $L_p$-masks which are ablation targets in (c). (b) Layer-wise distribution of learned $L_p$-mask properties. (c) Selectively ablation of $L_p$-masks near $90°$ by gradually increasing the ablation; Ablation in all 10 $L_p$-conv layers (left), first 5 layers (middle), and last 5 layers (right) respectively.

performance degradation compared to the baseline model. Notably, in contrast to the large kernel models, all instances of $L_p$-Conv, regardless of the value of $p_{\text{init}}$, exhibit stable performance gains, as indicated in Table 1. Intriguingly, the condition $p_{\text{init}} = 2$ (which is close to biological RFs) recorded the highest performances, ranking first in seven out of ten cases and second in two. This observation suggests that employing $L_p$-convolution to increase the kernel size can enhance the performance of historically successful CNN models as well as the recent models, particularly when $p_{\text{init}} = 2$, which aligns closely with biologically observed RFs. This underscores the potential benefits for CNNs from incorporating brain-inspired structured sparsity and flexible RF adaptability.

## 6 REPRESENTATIONAL SIMILARITY ANALYSIS BETWEEN $L_p$-CNNS AND VISUAL CORTEX

Figure 5a illustrates our approach to assessing the alignment of representations between biological and artificial systems. We utilized the standardized dataset from Allen Brain Observatory (de Vries et al., 2020), which uses 118 images of Natural Scenes and corresponding neural activities recorded from the mouse visual cortex (VC). Our method builds upon established Representational Similarity Analysis (RSA) techniques (Khaligh-Razavi & Kriegeskorte, 2014; Nguyen et al., 2020; Mehrer et al., 2020; Devereux et al., 2013; Diedrichsen & Kriegeskorte, 2017), to compare the representations of CNNs (Bakhtiari et al., 2021; Shi et al., 2019) (See details in Appendix A.11).

Table 1: Top-1 performance on the CIFAR-100 and TinyImageNet datasets in CNNs are reported with 5 trials (mean±std). The symbol † indicates that both $\mathcal{C}$ and $p$ are frozen parameters during training. $k' = 2 \times \lceil \frac{k}{2} \rceil + k$. For all $L_p$-Conv layers, $\mathcal{C}$ was initialized with $1/\sigma_{\text{init}}$ of diagonals and 0 of off-diagonals, where $\sigma_{\text{init}} = k/2$. Statistical comparison results using Welch's t-test with the base model are marked as follows: 'ns' (p-value $\geq 0.05$), '\*' ($0.01 \leq$ p-value $< 0.05$), '\*\*' ($0.001 \leq$ p-value $< 0.01$), and '\*\*\*' (p-value $< 0.001$). The text in bold denotes the best performance, while underlined signifies the second best. Gray indicates a baseline performance and red indicates a decrease in performance.

| | | **CIFAR-100** | | | | |
|---|---|---|---|---|---|---|
| Layer | $p_{\text{init}}$ | **AlexNet** | **VGG-16** | **ResNet-18** | **ResNet-34** | **ConvNeXt-T** |
| (Base) Conv | - | $66.05 \pm 0.33$ | $70.26 \pm 0.29$ | $71.22 \pm 0.18$ | $72.47 \pm 0.23$ | $58.36 \pm 6.48$ |
| (Large) Conv | - | ***$54.53 \pm 0.65$ | **$64.82 \pm 2.92$ | ***$72.80 \pm 0.27$ | ***$73.52 \pm 0.11$ | ns$54.13 \pm 1.14$ |
| †$L_p$-Conv | 256 | ns$65.95 \pm 0.32$ | **$71.03 \pm 0.38$ | ns$71.24 \pm 0.23$ | ns$72.61 \pm 0.27$ | ns$60.34 \pm 2.80$ |
| $L_p$-Conv | 16 | **$\mathbf{67.12 \pm 0.37}$ | **$70.87 \pm 0.23$ | ***$72.35 \pm 0.30$ | ***$73.32 \pm 0.23$ | ns$\underline{61.30 \pm 1.71}$ |
| $L_p$-Conv | 8 | **$66.85 \pm 0.18$ | **$71.14 \pm 0.29$ | ***$72.26 \pm 0.28$ | ***$73.37 \pm 0.15$ | ns$59.94 \pm 5.04$ |
| $L_p$-Conv | 4 | *$66.68 \pm 0.28$ | ***$\underline{71.71 \pm 0.36}$ | ***$73.00 \pm 0.15$ | ***$\underline{74.07 \pm 0.22}$ | ns$59.34 \pm 7.53$ |
| $L_p$-Conv | 2 | ns$66.13 \pm 0.33$ | ***$\mathbf{72.88 \pm 0.30}$ | ***$\mathbf{73.86 \pm 0.14}$ | ***$\mathbf{74.95 \pm 0.11}$ | ns$\mathbf{62.61 \pm 3.03}$ |
| | | **TinyImageNet** | | | | |
| Layer | $p_{\text{init}}$ | **AlexNet** | **VGG-16** | **ResNet-18** | **ResNet-34** | **ConvNeXt-T** |
| (Base) Conv | - | $52.25 \pm 0.35$ | $67.75 \pm 0.07$ | $66.63 \pm 0.51$ | $69.22 \pm 0.11$ | $70.25 \pm 0.45$ |
| (Large) Conv | - | ***$35.52 \pm 0.46$ | ns$66.96 \pm 1.50$ | ***$68.33 \pm 0.19$ | ns$69.46 \pm 0.36$ | ns$68.66 \pm 1.50$ |
| †$L_p$-Conv | 256 | ns$52.60 \pm 0.12$ | ns$67.72 \pm 0.18$ | ns$66.37 \pm 0.55$ | ns$69.27 \pm 0.27$ | ns$70.45 \pm 0.44$ |
| $L_p$-Conv | 16 | ***$53.98 \pm 0.50$ | ***$69.29 \pm 0.25$ | **$67.72 \pm 0.43$ | **$\underline{70.00 \pm 0.33}$ | ns$70.62 \pm 0.30$ |
| $L_p$-Conv | 8 | **$54.07 \pm 0.91$ | ***$69.72 \pm 0.16$ | *$67.63 \pm 0.45$ | ***$69.81 \pm 0.23$ | ns$70.52 \pm 0.36$ |
| $L_p$-Conv | 4 | ***$\mathbf{54.30 \pm 0.48}$ | ***$\underline{69.79 \pm 0.30}$ | **$68.20 \pm 0.50$ | **$69.99 \pm 0.44$ | ns$\mathbf{70.74 \pm 0.37}$ |
| $L_p$-Conv | 2 | ***$\underline{54.13 \pm 0.53}$ | ***$\mathbf{69.96 \pm 0.45}$ | **$\mathbf{68.45 \pm 0.36}$ | ***$\mathbf{70.43 \pm 0.24}$ | ns$\underline{70.72 \pm 0.31}$ |

Based on the observation that $L_p$-convolution tends to perform better as it approaches biologically observed RFs with $p_{\text{init}} = 2$, we posed the question of whether the neural representation of artificial models with biological RFs aligns more closely with the representation of VC. To address this, we compared the neural representations of TinyImageNet-trained CNNs from Table 1 by presenting the 118 Natural Scene images, with the mouse VC representations. To facilitate model comparison, we extracted the representative value, maximum SSM, chosen from pair-wise SSMs across the convolutional layers and the VC subregions (See pair-wise SSMs in Appendix A.12). In the results, models with $p_{\text{init}}$ closer to 2 generally exhibited better alignment with the brain (Fig. 5b). In summary, we find that $L_p$-convolution tends to achieve better alignment with the brain as it approximates a Gaussian distribution.

## 7    RELATED WORKS

**Resurgence of large kernels in convolution**    In the early stages of CNN development, large kernels were not widely adopted, with their use predominantly confined to the initial layers (Krizhevsky et al., 2012; Szegedy et al., 2015; 2017). Attempts to enlarge kernel size sometimes led to a decline in classification performance (Peng et al., 2017). Consequently, the more favored strategy was stacking smaller kernels (1x1, 3x3) (Simonyan & Zisserman, 2014; He et al., 2016).

The advent of Vision Transformers (ViTs) marked a paradigm shift from traditional CNN models, with the Swin Transformer emphasizing the significance of both attention mechanisms and large receptive fields, thereby renewing interest in large kernel CNNs (Dosovitskiy et al., 2020; Liu et al., 2021; Touvron et al., 2021; Vaswani et al., 2021; Liu et al., 2021). Recent innovations such as RepLKNet and SLaK have showcased performance comparable to ViTs, highlighting the potential of large kernel CNNs in modern computer vision (Ding et al., 2022; Liu et al., 2023). However, the effectiveness of large kernels in historically successful CNN models remains unexplored.

In this paper, we implemented large-kernel convolution by applying trainable masks to existing filters and introducing Gaussian-based structured sparsity for adjustments in receptive fields tailored to specific tasks. This methodology contrasts with recent approaches that incorporate unstructured row-rank sparsity directly into filters, which may necessitate extensive hyperparameter tuning. Our train-

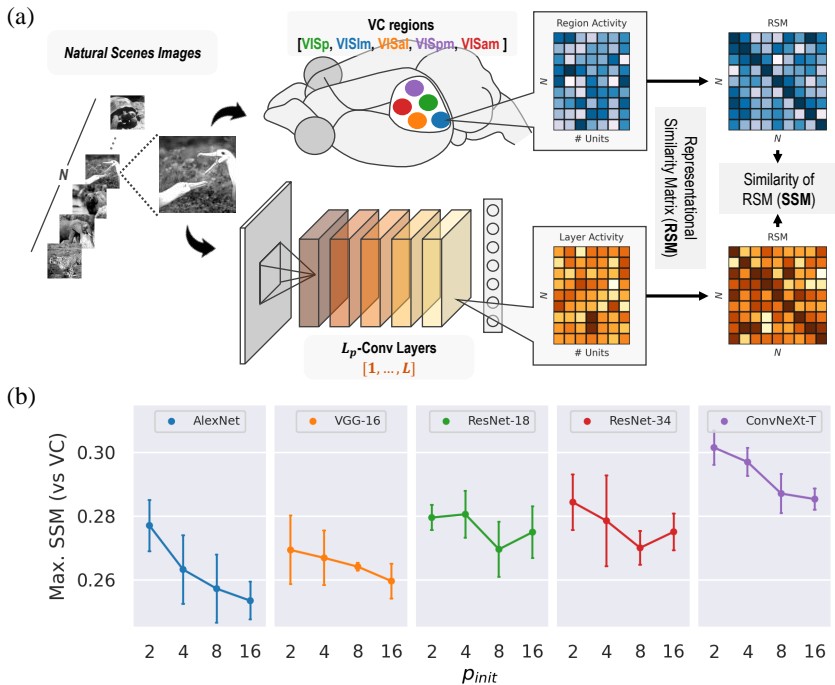

Figure 5: **Representational similarity between biological brain and artificial models using natural images** (a) Schematic illustration of representational similarity analysis from neural activities of mouse VC L2/3 subregions and convolutional layers of TinyImageNet-trained CNNs; Unit activities in both mouse brain or CNNs were obtained from $N$ number of image inputs; $N \times N$ representational similarity matrix (RSM) was constructed for every subregions or layers by measuring the correlations across unit activities. The similarity of the RSMs (SSM) between the V1 region and CNN Conv layer was measured by calculating Kendall's rank correlation coefficient. (b) For the comparison of the representational similarity score in $L_p$-models, the maximum SSM score was collected as the representative value among all pair-wise SSM scores across regions and layers; the two-sample Student's t-test was conducted for statistical analysis and demonstrated statistical significance ($p_{\text{init}} = 2$ vs. others) with p-value<0.05 (*), p-value<0.01 (**), p-value<0.001 (***).

able masks streamline the optimization process by automatically adjusting key parameters, thereby facilitating the application of large kernel training in both traditional and modern CNN architectures.

**Gaussian distribution as receptive field of visual information processing**   In biological systems, computational neuroimaging studies have uncovered that population receptive fields (pRFs) in the human visual system exhibit Gaussian distributions, shedding light on population-level insights (Wandell & Winawer, 2015). Recent research underscores the predominance of circular pRF shapes in the early visual cortex (Lerma-Usabiaga et al., 2021).

In the realm of artificial systems, the concept of effective receptive fields (ERFs) was introduced by Luo et al. (2016). Their findings indicate that ERFs in CNNs display Gaussian-like properties and are significantly smaller than their theoretical counterparts, thereby providing critical insights into how information is integrated across layers (Luo et al., 2016).

The prevailing understanding is that the Gaussian-like nature of RFs in both biological and artificial systems arises from the cumulative effect of convolutions through stacked layers. However, in our study, we show that the inherently Gaussian-like RFs — not a result of the cumulative stacking of layers —, could be beneficial in visual processing and more accurate in the explanation of the brain's neural representation.

## 8 CONCLUSION

The emergence of CNNs stands as one of the most significant events in the history of artificial intelligence (Krizhevsky et al., 2012). CNN-based models have led to the development of numerous commercial applications, generating substantial industrial value (Alzubaidi et al., 2021). Notably, CNNs have not only played a pivotal role in engineering but have also become a major model in the study of visual processing in the brain (Lindsay, 2021). However, with the recent advent of Vision Transformers, the throne of dominance as a vision model appears to be shifting away from CNNs (Dosovitskiy et al., 2020). Yet, the emergence of large-kernel CNNs has prompted a reevaluation of the potential of CNNs Ding et al. (2022); Liu et al. (2023).

In this study, we introduced a novel $L_p$-convolution, based on the MPND, with the objective of narrowing the gap between artificial and biological RFs, and subsequently crafting neural network modules more aligned with biological structures. Brain-inspired $L_p$-convolution enables the cultivation of diverse-shaped RFs with Gaussian-based structured sparsity, adaptable to various rotations, distortions, and scales, and tailored for specific tasks. Significantly, $L_p$-convolution showcases its adaptability and compatibility across an extensive spectrum of CNN models, from the conventional to the contemporary, underscoring its proficiency, especially in contexts involving large kernels. We believe our research serves as a noteworthy illustration of the symbiotic relationship between advancements in artificial intelligence and our understanding of neural processes.

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

# A APPENDIX

## A.1 BIOLOGICAL RFs

For the biological RF analysis, we have analyzed *in vivo* intracortical connectivity dataset of Rossi et al. (2020) collected from the mouse primary visual cortex (V1) (Fig. 1a). It contains both excitatory (CaMK2a-positive) and inhibitory (Gad2-positive) layer 2/3 neuronal spatial connectivity distribution (Fig. 1b) which was determined by recording GCaMP6 signals (calcium activities) of pre-and post-synaptic pairs (Fig. 7 and 6).

Given a post-synaptic neuron positioned at $(x_0, y_0)$, (Fig. 1b black), and the $N_b$ number of functional synapse positions $x_n, y_n \in (-\infty, \infty)$ for $n = 1, 2, ..., N_b$, then relative offsets are defined as $(\Delta x_n, \Delta y_n) = (x_n - x_0, y_n - y_0)$. We summarize functional synapse positions for biological RF as following

$$\Delta \mathbf{s_b} = [(\Delta x_n, \Delta y_n)]_{n=1}^{N_b}. \tag{5}$$

## A.2 ARTIFICIAL RFs

For the artificial RF analysis, we used untrained or pre-trained AlexNet[4] with inputs ($224 \times 224$) of 17 images either generated from Gaussian noises or selected among 118 Natural Scenes images datasets (See details in Appendix A.10 and Fig. 7). When image inputs were shown to AlexNet, we extracted RFs of the functional synapse from the first convolutional layer (Conv1) (Fig. 1, c and d).

Given the input $\mathcal{X} \in \mathbb{R}^{C_i \times H \times W}$ and weights parameters for Artificial RFs $\mathcal{W}_{\text{ARF}} \in \mathbb{R}^{C_i \times K_h \times K_w}$, the post-synaptic unit in the Convolution layer, receives weighted-input $\mathcal{Z} \in \mathbb{R}^{V \times C_i \times K_h \times K_w}$, the results of element-wise multiplication between partial input and filters, where $\mathcal{Z}_{v=m*(H-K_h+1)+n} = \mathcal{X}_{\cdot,m:m+K_h-1,n:n+K_w-1} \odot \mathcal{W}$ (Fig. 1d black) for $0 \leq m \leq H - K_h$ and $\leq n \leq W - K_w$. We calculate the weighted input as the convolution operation without summation across width, height and input channel. As a result, weighted input $\mathcal{Z}$ has $V \times C_i \times K_h \times K_w$ shape, where $V = (H - K_h + 1) \times (W - K_w + 1)$, and $\mathcal{Z}_v \in \mathbb{R}^{C_i \times K_h \times K_w}$ denotes the $v$-th element of $\mathcal{Z}$. Here, $C_i$, $H$, $W$, $K_h$ and $K_w$ denote the number of input channels, input height, input width, kernel height and kernel width, respectively. For simplicity, we assume that there is stride one and no zero-padding. For $h \in [0, \cdots, K_h - 1]$ and $w \in [0, \cdots, K_w - 1]$, the relative offsets from the kernel center are defined as follows

$$\Delta s = (\Delta h, \Delta w) = (h - \frac{K_h}{2}, w - \frac{K_w}{2}). \tag{6}$$

Since spatial connectivity pattern in the biological synapse is measured by the functional calcium activities and given as coordinates, we applied a similar approach to that of CNN layers. We collected $N_a$ functional weighted-inputs (functional synapses) where $N_a$ represents the number of cases where each elements of $|\mathcal{Z}|$ exceeds a threshold $\theta$. Here, we defined $\theta$ as the standard deviation of $|\mathcal{Z}|$ [5]. This selection process yielded a different set of functional synapses input-dependent manner. We summarize functional synapse positions for artificial RF as following

$$\Delta \mathbf{s_a} = \{(v, k, \Delta h, \Delta w) | \mathcal{Z}_{v,k,h,w} > \theta\}, \text{ where } |\Delta \mathbf{s_a}| = N_a. \tag{7}$$

---

[4]For the pre-trained model, we used the torchvision's ImageNet-1k pre-trained model

[5]We determined the activity threshold based on a common method used in neuroscience to extract meaningful patterns in neural activity, which is similar to calculating the Z-score and typically setting a threshold at a range of 2 to 3 standard deviations to identify values that are statistically significant.

### A.3 GAUSSIAN DISTRIBUTED FUNCTIONAL SYNAPSES OF POST-SYNAPTIC NEURON IN MOUSE V1 LAYER 2/3

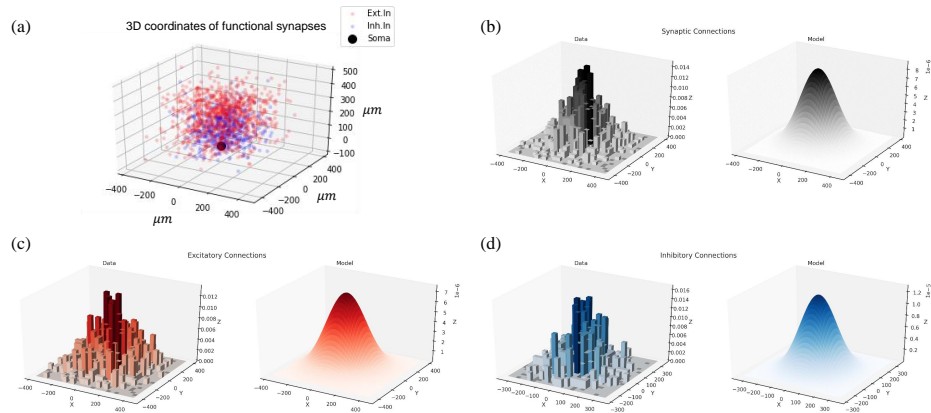

Figure 6: **Distribution of post-synaptic functional synapses in mouse V1 layer 2/3** (a) Using dataset from Rossi et al. (2020), 3D scatter plot represent relative positions of both excitatory (red) and inhibitory (blue) functional synapses from the soma of the post-synaptic neuron. (b-d) 2d histogram (left) and Gaussian fitted probability density function (right), showing the laminar organization of functional synapses for all (b), excitatory (c), and inhibitory (d)

### A.4 INDIVIDUAL RECEPTIVE FIELDS COLLECTED FROM BOTH BIOLOGICAL AND ARTIFICIAL SYSTEMS

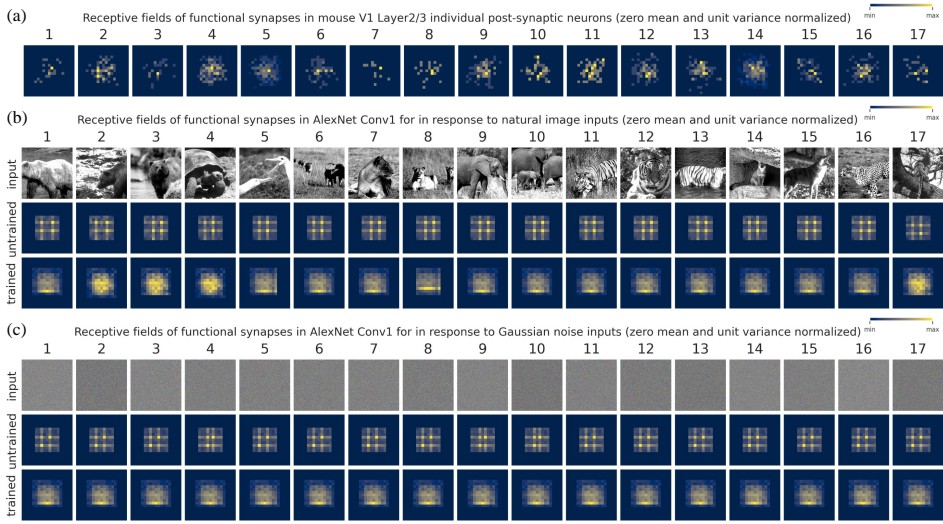

Figure 7: **Biological and artificial receptive fields with visual stimulus** The receptive field discussed in this figure specifically refers to the spatial connectivity patterns of synapses. Note that this differs from receptive fields typically associated with low-level visual feature selectivity. (a) Receptive fields of individual neurons in V1 Layer2/3 from the dataset Rossi et al. (2020) (b) Receptive fields of untrained or ImageNet-1k pretrained AlexNet's Conv1 layer when Natural Scenes images were shown (c) Receptive fields of untrained or ImageNet-1k pretrained AlexNet's Conv1 layer when Gaussian RGB noise were shown. All receptive fields are zero mean unit variance normalized.

## A.5 EXPERIMENTAL DETAILS FOR SUDOKU CHALLENGE

We utilized the extensive 1M-sudoku dataset (Park, 2018), a resource also utilized in prior works (Amos & Kolter, 2017; Palm et al., 2018; Wang et al., 2019). Sudoku, a widely popular number puzzle, involves organizing digits in a grid such that each row (1×9), column (9×1), and box (3×3) contains all numbers from 1 to 9. In the Sudoku challenge, where achieving these three objectives simultaneously is essential for complete Sudoku solving, there is an advantage that we can test the applicability and effectiveness of evolving RFs in the $L_p$-convolution. To achieve this, we compared five distinct models of Sudoku CNN: (3×3) Base model, (7×7) Large model, (7×7) $L_p(p_{\text{init}} = 2)$, (7×7) $L_p(p_{\text{init}} = 16)$, and finally (7×7) $L_p^{\dagger}(p_{\text{init}} = 256)$ (frozen $p$ and $\mathbf{C}$ model) 2. For numerical stability, we clipped $p \geq 2$ during the training of the Sudoku-solving task. The inputs, targets, and model architecture are outlined in Figure 4a. As illustrated, our model comprises repeated Conv2dSame blocks, originally introduced in SudokuCNN (Oinar, 2021). Each Conv2dSame block encompasses Reflection padding, followed by a conventional Convolutional or $L_p$-Convolutional layer, Batch Normalization, and an activation function. The Convolutional layer has 256 channels, and the number of blocks is set at $L = 10$.

## A.6 CONFORMATIONAL ANALYSIS OF $L_p$-MASKS

We defined the properties of scale ($\alpha$), rotation ($\theta$), and distortion ($\gamma$) through Singular Value Decomposition (SVD) on $\mathbf{C}$, covariance matrix for each output channel, as shown in the following equation:

$$\mathbf{C} = \mathbf{U}\mathbf{\Lambda}\mathbf{V}^T. \tag{8}$$

Here, $\mathbf{U}$ and $\mathbf{V}$ represent $2 \times 2$ unitary matrices containing the left and right singular vectors. $\mathbf{\Lambda}$ is a diagonal matrix containing the singular values $(\lambda_0, \lambda_1)$. Rotation is quantified as $\theta = \arctan\left(\frac{\sin(\mathbf{V}^T[1])}{\cos(\mathbf{V}^T[1])}\right)$ (in degrees), providing a measure of rotational transformation. Distortion is quantified as $\gamma = \frac{\lambda_0}{\lambda_1}$, offering valuable information about the deformation present in the data. Scale is quantified as $\alpha = \sqrt{\left(\frac{1}{\lambda_0}\right)^2 + \left(\frac{1}{\lambda_1}\right)^2}$, indicating the size of mask. In Figure 4a, we show the example shapes of $L_p$-masks by reverse calculating $\mathbf{C}$ from the given $\alpha, \theta, \gamma$.

## A.7 LAYER-WISE VISUALIZATION OF $L_p$-MASKS FOR SUDOKU-$L_p$CONVNET

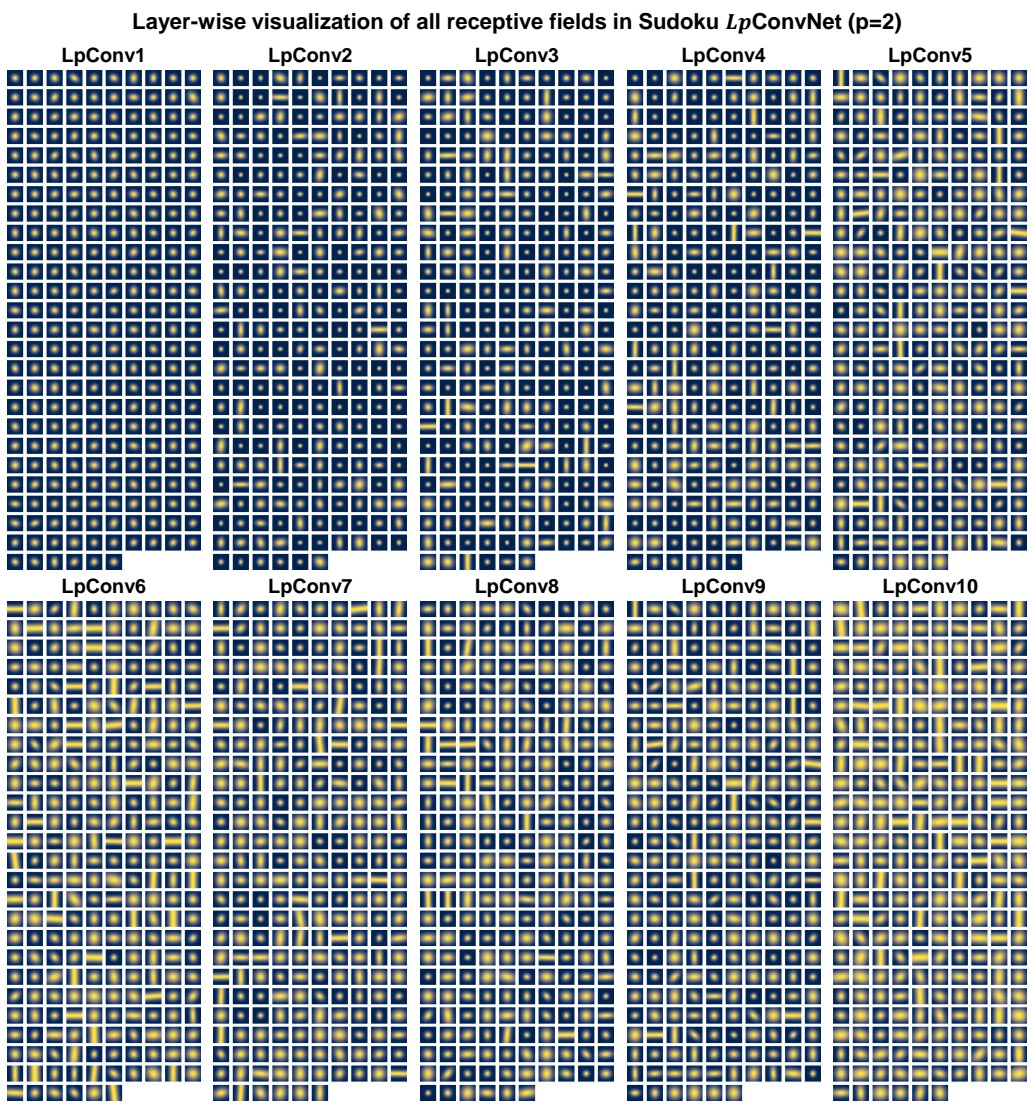

Figure 8: **Layer-wise visualization of $L_p$-masks for Sudoku-$L_p$convNet** All learned $L_p$-masks after Sudoku task training of $L_p$-ConvNet($p_{\text{init}} = 2$). With an increase in layer depth, the sizes of masks get larger.

A.8  ACCURACY CURVES FOR SUDOKU ABLATION EXPERIMENTS ON $L_p$(P=2)

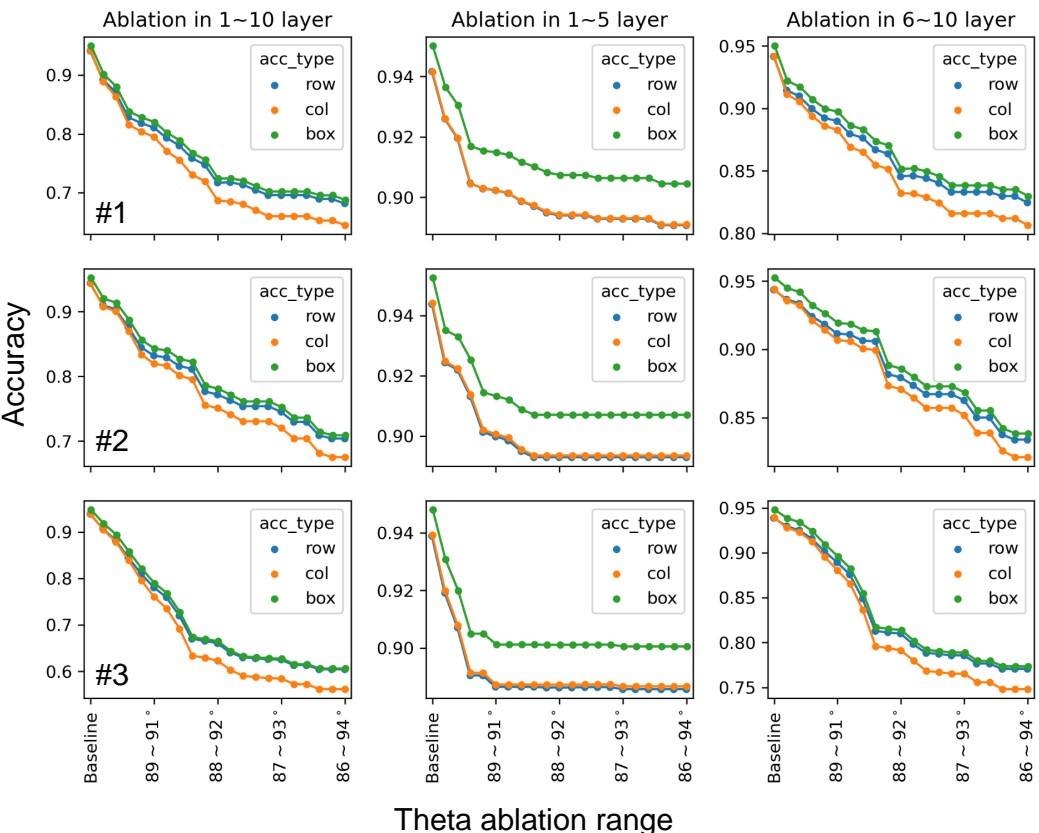

Figure 9: **Accuracy curves for Sudoku ablation experiments on $L_p$(p=2)**. 3 individual experiments (rows) of vertical $L_p$-masks ablations with 3 different conditions (columns; layer 1 to 10, left; layer 1 to 5, middle; layer 6 to 10, right) on $L_p$(p=2)

## A.9 EXPERIMENTAL DETAILS OF VISION CLASSIFICATION TASK

We conducted our training on two datasets: CIFAR-100 (Krizhevsky et al., 2009) and TinyImageNet (Le & Mikolov, 2014). CIFAR-100 comprises $32 \times 32$ pixel images distributed across 100 classes, while TinyImageNet consists of $224 \times 224$ pixel images spanning 200 categories. Following standard procedures, we reported top-1 accuracy with corresponding mean and standard deviation. Our implementation is based on the **PyTorch** framework (Paszke et al., 2019), making extensive use of the **timm** repository (Wightman, 2019). We adopted a training strategy rooted in DeiT (Touvron et al., 2021), incorporating techniques such as RandAugment (Cubuk et al.), Mixup (Zhang et al., 2017), Cutmix (Yun et al., 2019), random erasing (Zhong et al., 2020), and stochastic depth (Huang et al., 2016). The optimization process employed AdamW (Loshchilov & Hutter, 2017) with a default momentum value of 0.9 and a weight decay set at $5 \times 10^{-2}$. We initialized our learning rate at $1 \times 10^{-3}$ and implemented a cosine learning rate schedule. All models underwent training for 300 epochs, utilizing a batch size of 128. For CIFAR-100, training was conducted on 2 GTX 1080ti GPUs, while 2 Tesla V100 GPUs were used for TinyImageNet.

## A.10 THE ALLEN BRAIN OBSERVATORY DATASET

The Allen Brain Observatory dataset (de Vries et al., 2020) constitutes a comprehensive standardized in vivo examination of physiological activity within the mouse visual cortex. It encompasses recordings of visually-induced calcium responses from neurons expressing GCaMP6f. This dataset encompasses cortical activity from nearly 60,000 neurons originating from six distinct visual areas, four layers, and twelve transgenic mouse Cre lines. These recordings were gathered from 243 adult mice in reaction to a diverse set of visual stimuli. In this study, we focused on utilizing the collective neural responses from five visual areas (VISal, VISam, VISl, VISp, VISpm), Layer 2/3 (depth range of 175mm to 275mm), and three mouse lines (Cux2-CreERT, Emx1-IRES-Cre, Slc17a7-IRES2-Cre) when presenting a dataset of natural scenes to the mice. This dataset comprised 118 natural images obtained from three different databases (Berkeley Segmentation Dataset (Martin et al., 2001), van Hateren Natural Image Dataset (Van Hateren & van der Schaaf, 1998), and McGill Calibrated Colour Image Database (Olmos & Kingdom, 2004)). Further details regarding the experiment can be found in (de Vries et al., 2020). In our study, we employed both images and neural responses for experiments involving representational similarity analysis to evaluate the correspondence between CNNs and the visual cortex, mirroring earlier investigations (Shi et al., 2019; Bakhtiari et al., 2021).

## A.11 REPRESENTATION SIMILARITY ANALYSIS

While the details of RSA are expertly addressed in Diedrichsen & Kriegeskorte (2017), let us briefly cover our specific approach. We leveraged the codebase provided by Bakhtiari et al. (2021). In RSA, we generate response matrices ($\mathbf{R}$) for brain regions and neural network layers, with dimensions $N \times M$ (where $N$ is the number of image inputs and $M$ is the neuron count). Using Pearson correlation, we compute similarities within matrix $\mathbf{R}$ to construct the $N \times N$ Representation Similarity Matrix (RSM). Additionally, following the methodology of Bakhtiari et al. (2021), we applied noise correction by normalizing the RSAs using the noise ceiling values. These values were obtained through comparisons of representations across different mice. For example, the noise ceiling value for VISp is derived by calculating the RSMs of VISp from different animals and taking their median. To assess the similarity between RSMs (SSM), we employ Kendall's $\tau$ for robust agreement, which helps mitigate potential bias from measurement noise Diedrichsen et al. (2020).

### A.12 PAIR-WISE REPRESENTATION SIMILARITY ANALYSIS BETWEEN ALL CNN LAYERS AND V1 SUBREGIONS

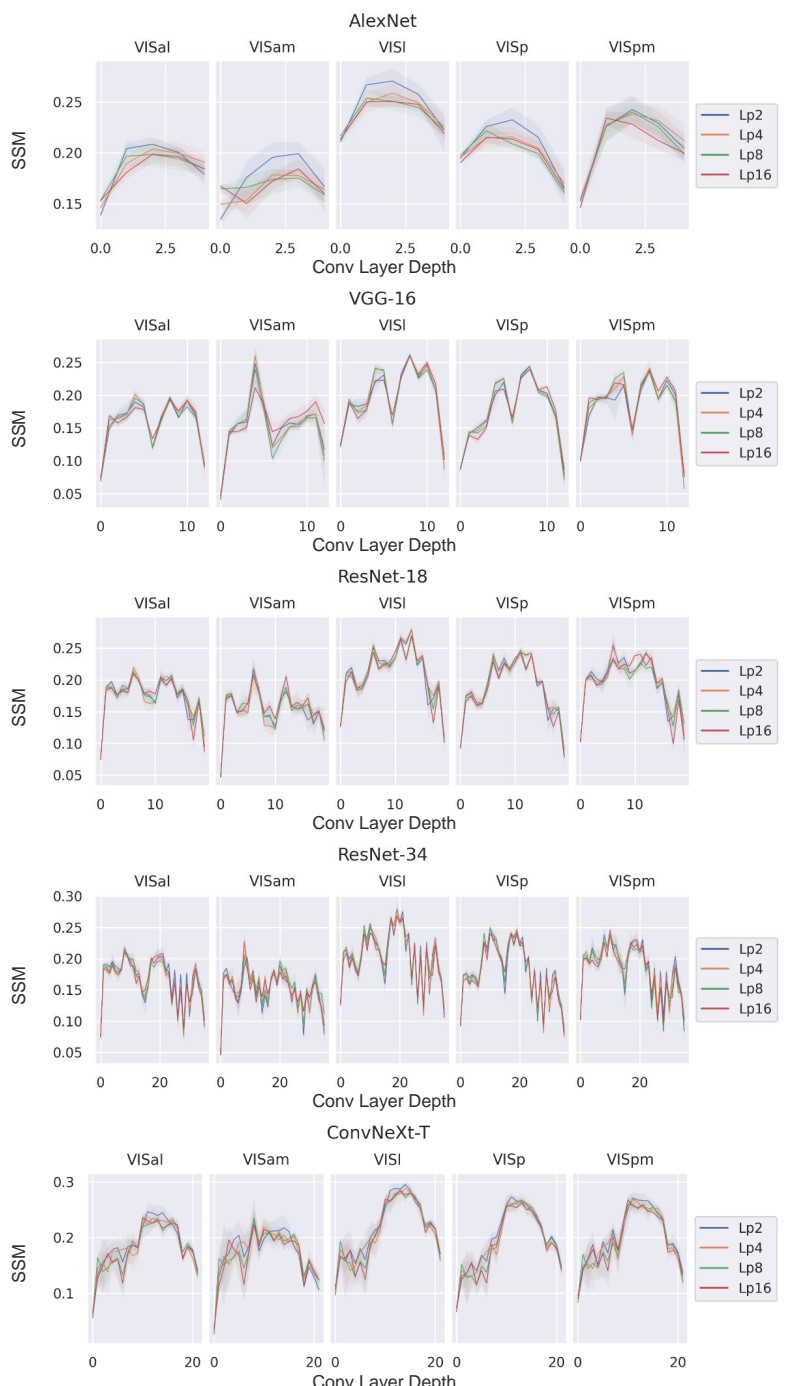

Figure 10: **Pair-wise representation similarity analysis between all CNN layers and VC subregions**. We show the SSM scores for all pairs of Conv layers from CNNs and VC subregions. y-axis, SSM score; x-axis, Conv layer depth. For Max. SSM, we choose the highest SSM among all pair-wise SSMs.

## A.13 PyTorch-style pseudocode for $L_p$-Convolution

**Algorithm 1:** PyTorch-style pseudocode for $L_p$-Convolution

```python
import torch
import torch.nn as nn
import torch.nn.functional as F

from torchvision.models import alexnet

class LpConv2d(nn.Conv2d):
    def __init__(self, p_init, sigma_init, in_channels,
                 out_channels, kernel_size, stride,
                 padding, **kwargs):

        ...

        # Create parameters p & C
        params_p = torch.ones( out_channels ) * p_init
        params_C = torch.zeros( out_channels, 2, 2 ) )
        params_C[:,0,0] = 1/sigma_init
        params_C[:,1,1] = 1/sigma_init

        self.p = nn.Parameter( params_p )
        self.C = nn.Parameter( params_C )

    def forward(self, input):
        # Create channel-wise lp_masks from parameters p and C
        lp_masks = get_channel_wise_lp_masks(self.p, self.C)

        # Overlay lp_masks on weight
        masked_weight = weight * lp_masks
        return F.conv2d(inputs, masked_weight, bias,
                        kernel_size, stride, padding, **kwargs)

def LpConvert(model, p_init):
    # Convert all nn.Conv2d layers into LpConv2d
    for i in range(num_layers):
        layer = model.layers[i]
        if layer is nn.Conv2d:
            model.layers[i] = LpConv2d(p_init, sigma_init, **kwargs)
    return model

# LpConvert on Alexnet for TinyImageNet
base_model = alexnet(num_classes=200)
lp2_model = LpConvert(base_model, p_init=2)
```

A.14 VISUALIZATION OF FUNCTIONAL RECEPTIVE FIELDS OF PRE-TRAINED ALEXNET

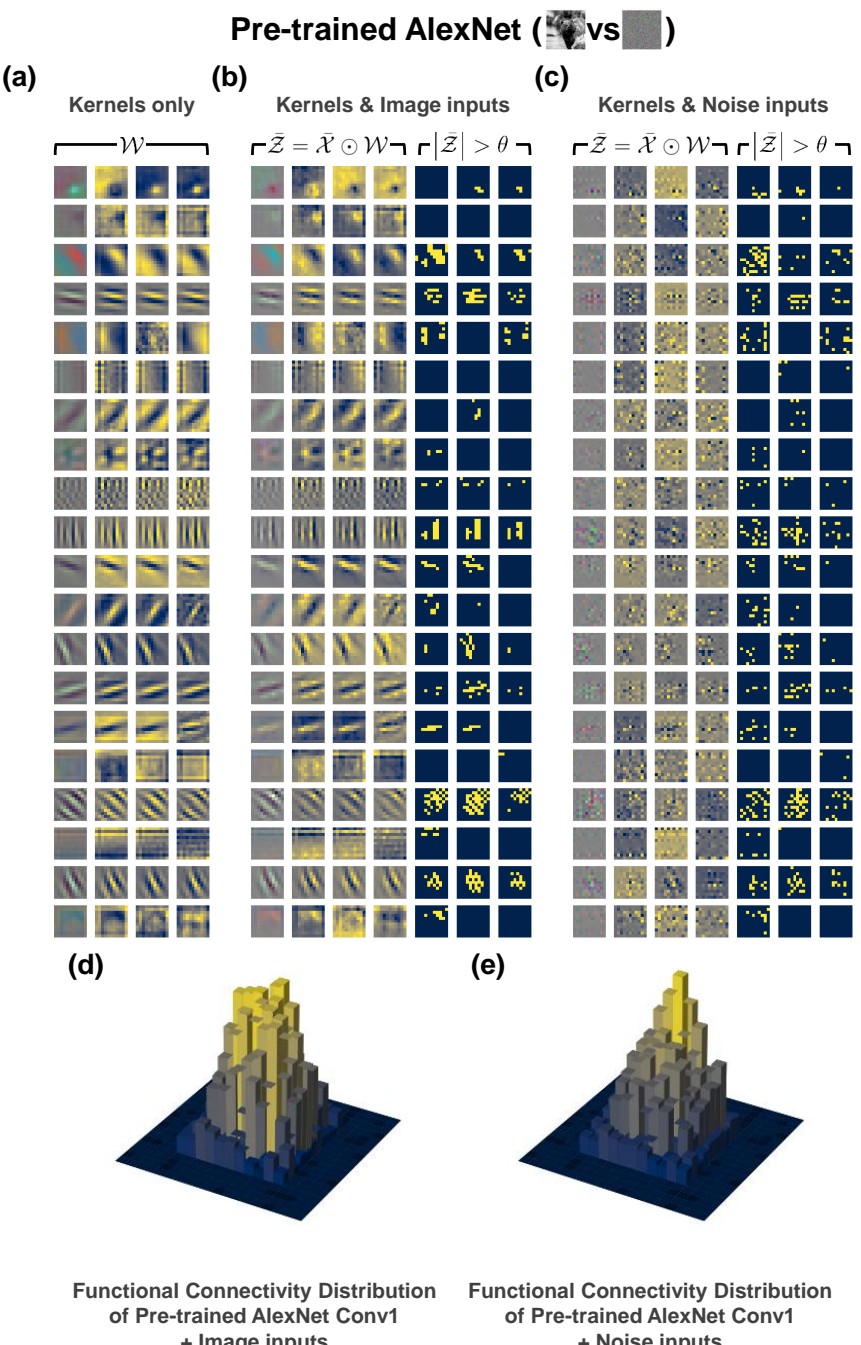

Figure 11: **Visualization of functional receptive fields of pre-trained AlexNet Conv1 with image or noise inputs**. Visualization of first 20 kernels of total 64 without inputs (a; column orders: RGB, R, G, B), with image inputs (b; column orders: RGB, R, G, B, R, G, B) with noise inputs (c; column orders: RGB, R, G, B, R, G, B). (d) Histogram of functional connectivity from (b). (e) Histogram of functional connectivity from (c). $\mathcal{W}$, Weight; $\bar{\mathcal{X}}$, kernel-sized input; $\bar{\mathcal{Z}}$, kernel-sized output; $\theta$, activity threshold; $\odot$, element-wise product. See Appendix A.2. for methodological details.

### A.15 VISUALIZATION OF FUNCTIONAL RECEPTIVE FIELDS OF UNTRAINED ALEXNET

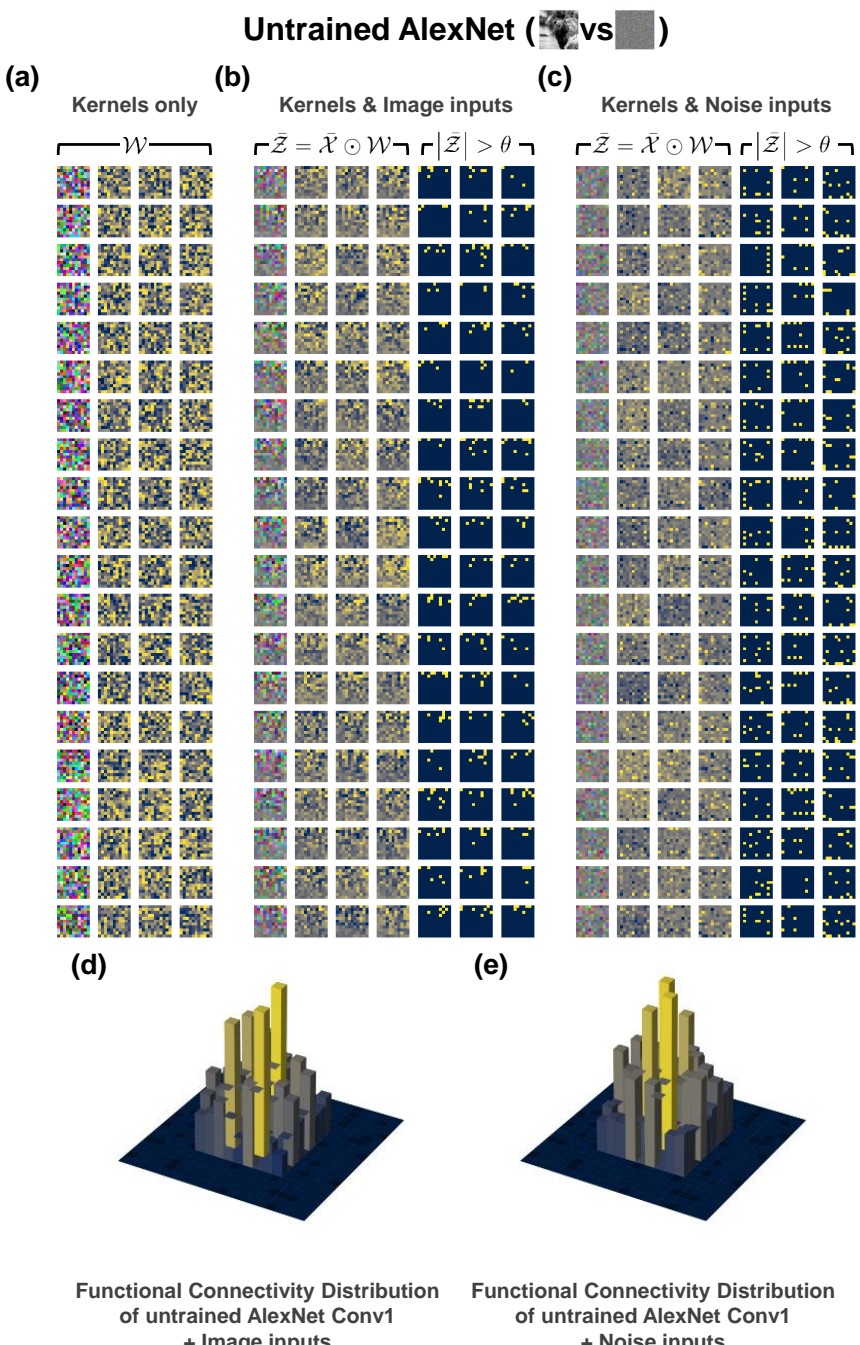

Figure 12: **Visualization of functional receptive fields of untrained AlexNet Conv1 with image or noise inputs**. Visualization of first 20 kernels of total 64 without inputs (a; column orders: RGB, R, G, B), with image inputs (b; column orders: RGB, R, G, B, R, G, B) with noise inputs (c; column orders: RGB, R, G, B, R, G, B). (d) Histogram of functional connectivity from (b). (e) Histogram of functional connectivity from (c). $\mathcal{W}$, Weight; $\bar{\mathcal{X}}$, kernel-sized input; $\bar{\mathcal{Z}}$, kernel-sized output; $\theta$, activity threshold; $\odot$, element-wise product. See Appendix A.2. for methodological details.

### A.16 Visualization of MPND when $p < 2$

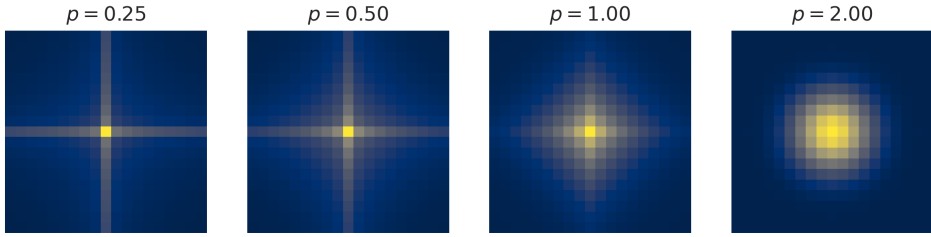

Figure 13: **Visualization of MPND when** $p < 2$ Given the value of $p = 1$, MPND distribution becomes diamond shape. When $p < 1$, the distribution becomes a cross-like shape.

### A.17 Post-trained $p$-distribution in $L_p$-masks

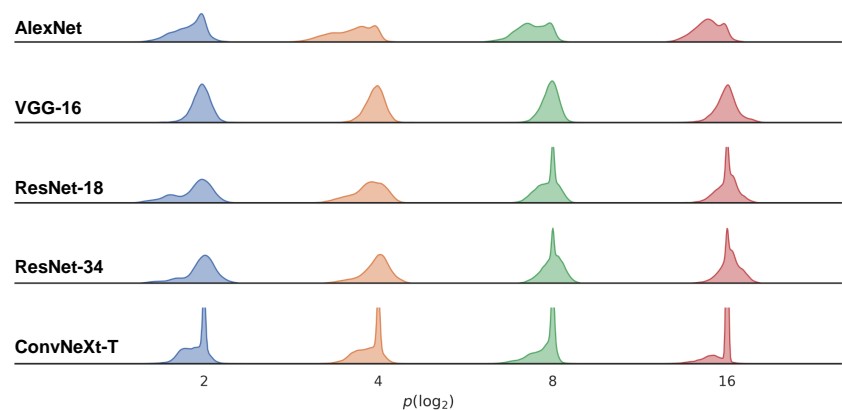

Figure 14: **CIFAR-100-trained $p$-distribution of $L_p$-masks**

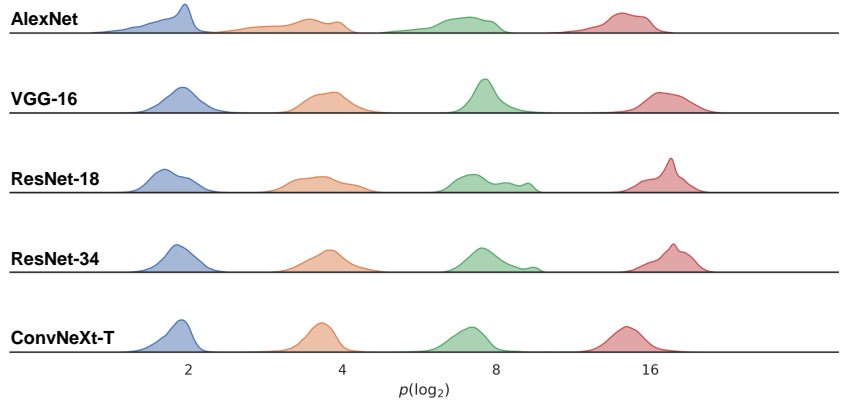

Figure 15: **TinyImageNet-trained $p$-distribution of $L_p$-masks**

Table 2: **Layer-wise $p$-distribution of CIFAR-100-trained $L_p$-masks** All values (Median $\pm$ Stdev) are calculated with $p$ of all $L_p$-masks in each layer, from 5 different trials of CIFAR-100-trained models.

**CIFAR-100**

| Model | Layer | $L_p$-Conv (p=2) | $L_p$-Conv (p=4) | $L_p$-Conv (p=8) | $L_p$-Conv (p=16) |
|---|---|---|---|---|---|
| AlexNet | 1 | 1.99 ± 0.08 | 3.88 ± 0.16 | 7.75 ± 0.35 | 15.45 ± 0.68 |
| | 2 | 2.00 ± 0.06 | 3.93 ± 0.19 | 7.67 ± 0.52 | 15.35 ± 0.96 |
| | 3 | 1.91 ± 0.07 | 3.70 ± 0.16 | 7.64 ± 0.37 | 15.06 ± 0.79 |
| | 4 | 1.80 ± 0.13 | 3.38 ± 0.32 | 7.18 ± 0.36 | 14.73 ± 0.64 |
| | 5 | 1.82 ± 0.11 | 3.35 ± 0.29 | 7.11 ± 0.38 | 14.45 ± 0.68 |
| ResNet-18 | 1 | 1.98 ± 0.11 | 3.85 ± 0.23 | 7.70 ± 0.49 | 15.39 ± 1.27 |
| | 2 | 2.01 ± 0.06 | 3.87 ± 0.13 | 7.91 ± 0.32 | 15.37 ± 0.84 |
| | 3 | 1.93 ± 0.05 | 3.68 ± 0.11 | 7.45 ± 0.25 | 15.30 ± 0.58 |
| | 4 | 1.95 ± 0.05 | 3.72 ± 0.12 | 7.61 ± 0.26 | 15.43 ± 0.61 |
| | 5 | 1.92 ± 0.05 | 3.65 ± 0.12 | 7.45 ± 0.23 | 15.55 ± 0.52 |
| | 6 | 1.98 ± 0.05 | 3.80 ± 0.10 | 7.58 ± 0.20 | 16.08 ± 0.56 |
| | 7 | 1.82 ± 0.05 | 3.45 ± 0.13 | 7.30 ± 0.26 | 14.92 ± 0.57 |
| | 8 | 1.71 ± 0.06 | 3.57 ± 0.17 | 8.00 ± 0.01 | 16.00 ± 0.00 |
| | 9 | 1.88 ± 0.05 | 3.61 ± 0.12 | 7.49 ± 0.21 | 15.47 ± 0.58 |
| | 10 | 1.93 ± 0.05 | 3.70 ± 0.11 | 7.50 ± 0.22 | 15.82 ± 0.48 |
| | 11 | 2.01 ± 0.06 | 3.90 ± 0.13 | 7.68 ± 0.21 | 16.28 ± 0.56 |
| | 12 | 1.79 ± 0.06 | 3.52 ± 0.13 | 7.35 ± 0.20 | 15.31 ± 0.51 |
| | 13 | 1.62 ± 0.05 | 3.42 ± 0.14 | 8.00 ± 0.01 | 16.00 ± 0.00 |
| | 14 | 1.93 ± 0.06 | 3.82 ± 0.11 | 7.72 ± 0.20 | 16.22 ± 0.56 |
| | 15 | 2.01 ± 0.05 | 3.96 ± 0.10 | 7.93 ± 0.20 | 16.54 ± 0.54 |
| | 16 | 2.09 ± 0.06 | 4.16 ± 0.11 | 8.07 ± 0.21 | 16.80 ± 0.60 |
| | 17 | 1.94 ± 0.05 | 3.84 ± 0.10 | 7.69 ± 0.22 | 15.54 ± 0.41 |
| | 18 | 1.74 ± 0.04 | 3.86 ± 0.12 | 8.00 ± 0.00 | 16.00 ± 0.00 |
| | 19 | 1.99 ± 0.06 | 4.02 ± 0.11 | 8.10 ± 0.21 | 16.19 ± 0.35 |
| | 20 | 2.02 ± 0.05 | 4.09 ± 0.09 | 8.21 ± 0.18 | 16.30 ± 0.25 |
| ResNet-34 | 1 | 1.99 ± 0.12 | 3.87 ± 0.26 | 7.73 ± 0.44 | 15.51 ± 1.01 |
| | 2 | 2.01 ± 0.06 | 3.87 ± 0.12 | 7.99 ± 0.33 | 15.25 ± 0.85 |
| | 3 | 1.96 ± 0.05 | 3.75 ± 0.10 | 7.65 ± 0.28 | 15.35 ± 0.51 |
| | 4 | 1.96 ± 0.05 | 3.78 ± 0.11 | 7.83 ± 0.25 | 15.54 ± 0.58 |
| | 5 | 1.93 ± 0.05 | 3.71 ± 0.12 | 7.56 ± 0.26 | 15.56 ± 0.49 |
| | 6 | 1.96 ± 0.05 | 3.76 ± 0.11 | 7.65 ± 0.23 | 15.72 ± 0.57 |
| | 7 | 1.94 ± 0.05 | 3.74 ± 0.11 | 7.58 ± 0.24 | 15.80 ± 0.45 |
| | 8 | 2.01 ± 0.05 | 3.90 ± 0.11 | 7.72 ± 0.21 | 16.27 ± 0.52 |
| | 9 | 1.83 ± 0.05 | 3.52 ± 0.11 | 7.45 ± 0.28 | 14.83 ± 0.58 |
| | 10 | 1.69 ± 0.06 | 3.49 ± 0.17 | 8.00 ± 0.01 | 16.00 ± 0.00 |
| | 11 | 1.89 ± 0.05 | 3.69 ± 0.11 | 7.61 ± 0.22 | 15.60 ± 0.52 |
| | 12 | 1.93 ± 0.05 | 3.76 ± 0.12 | 7.69 ± 0.25 | 15.81 ± 0.49 |
| | 13 | 1.95 ± 0.05 | 3.79 ± 0.10 | 7.72 ± 0.21 | 15.95 ± 0.52 |
| | 14 | 1.99 ± 0.05 | 3.89 ± 0.12 | 7.85 ± 0.24 | 16.16 ± 0.52 |
| | 15 | 1.97 ± 0.05 | 3.85 ± 0.11 | 7.77 ± 0.21 | 16.06 ± 0.52 |
| | 16 | 2.03 ± 0.05 | 3.98 ± 0.11 | 7.98 ± 0.23 | 16.39 ± 0.52 |
| | 17 | 2.06 ± 0.06 | 4.07 ± 0.12 | 7.91 ± 0.23 | 16.67 ± 0.51 |
| | 18 | 1.80 ± 0.07 | 3.59 ± 0.12 | 7.41 ± 0.21 | 15.41 ± 0.56 |
| | 19 | 1.65 ± 0.05 | 3.48 ± 0.14 | 8.00 ± 0.01 | 16.00 ± 0.00 |
| | 20 | 1.93 ± 0.06 | 3.87 ± 0.11 | 7.78 ± 0.21 | 16.29 ± 0.51 |
| | 21 | 2.01 ± 0.06 | 4.03 ± 0.10 | 8.04 ± 0.21 | 16.45 ± 0.53 |
| | 22 | 2.00 ± 0.06 | 4.00 ± 0.10 | 8.01 ± 0.20 | 16.49 ± 0.56 |
| | 23 | 2.05 ± 0.06 | 4.10 ± 0.10 | 8.19 ± 0.21 | 16.59 ± 0.52 |
| | 24 | 2.04 ± 0.05 | 4.06 ± 0.10 | 8.12 ± 0.20 | 16.57 ± 0.55 |
| | 25 | 2.03 ± 0.06 | 4.11 ± 0.12 | 8.23 ± 0.24 | 16.83 ± 0.51 |
| | 26 | 2.05 ± 0.05 | 4.08 ± 0.10 | 8.15 ± 0.20 | 16.59 ± 0.53 |
| | 27 | 2.02 ± 0.07 | 4.09 ± 0.13 | 8.24 ± 0.27 | 16.89 ± 0.52 |
| | 28 | 2.05 ± 0.06 | 4.08 ± 0.11 | 8.14 ± 0.20 | 16.51 ± 0.52 |
| | 29 | 2.02 ± 0.06 | 4.11 ± 0.13 | 8.32 ± 0.28 | 16.86 ± 0.50 |
| | 30 | 2.16 ± 0.06 | 4.31 ± 0.10 | 8.39 ± 0.20 | 17.12 ± 0.61 |
| | 31 | 1.95 ± 0.05 | 3.86 ± 0.09 | 7.75 ± 0.19 | 15.63 ± 0.37 |
| | 32 | 1.79 ± 0.04 | 3.95 ± 0.11 | 8.00 ± 0.00 | 16.00 ± 0.00 |
| | 33 | 1.98 ± 0.05 | 3.99 ± 0.10 | 8.05 ± 0.19 | 16.12 ± 0.29 |
| | 34 | 2.02 ± 0.04 | 4.06 ± 0.08 | 8.16 ± 0.18 | 16.26 ± 0.26 |
| | 35 | 1.97 ± 0.06 | 3.98 ± 0.11 | 8.01 ± 0.21 | 16.01 ± 0.27 |
| | 36 | 2.07 ± 0.05 | 4.14 ± 0.10 | 8.25 ± 0.30 | 16.22 ± 0.45 |
| VGG-16 | 1 | 2.07 ± 0.09 | 4.08 ± 0.17 | 8.22 ± 0.37 | 16.62 ± 0.85 |
| | 2 | 2.00 ± 0.06 | 3.90 ± 0.12 | 7.77 ± 0.25 | 15.88 ± 0.63 |
| | 3 | 1.98 ± 0.05 | 3.88 ± 0.13 | 7.95 ± 0.31 | 15.51 ± 0.80 |
| | 4 | 1.98 ± 0.06 | 3.83 ± 0.13 | 7.69 ± 0.26 | 15.60 ± 0.60 |
| | 5 | 1.96 ± 0.06 | 3.86 ± 0.13 | 7.94 ± 0.26 | 15.43 ± 0.68 |
| | 6 | 1.94 ± 0.07 | 3.82 ± 0.13 | 7.80 ± 0.25 | 15.87 ± 0.55 |
| | 7 | 1.94 ± 0.06 | 3.81 ± 0.12 | 7.69 ± 0.22 | 15.94 ± 0.58 |
| | 8 | 1.98 ± 0.06 | 3.93 ± 0.11 | 7.94 ± 0.26 | 15.94 ± 0.69 |
| | 9 | 2.02 ± 0.06 | 4.04 ± 0.10 | 8.07 ± 0.24 | 16.17 ± 0.67 |
| | 10 | 1.98 ± 0.07 | 3.98 ± 0.12 | 7.91 ± 0.18 | 16.59 ± 0.90 |
| | 11 | 1.91 ± 0.08 | 3.88 ± 0.12 | 7.80 ± 0.23 | 15.78 ± 0.44 |
| | 12 | 2.01 ± 0.06 | 4.06 ± 0.11 | 8.09 ± 0.20 | 16.19 ± 0.35 |
| | 13 | 2.00 ± 0.05 | 4.01 ± 0.09 | 8.00 ± 0.20 | 15.95 ± 0.43 |
| ConvNeXt-T | 1 | 2.06 ± 0.11 | 3.99 ± 0.15 | 7.97 ± 0.39 | 15.71 ± 0.95 |
| | 2 | 1.88 ± 0.06 | 3.66 ± 0.15 | 7.09 ± 0.34 | 14.47 ± 0.55 |
| | 3 | 1.86 ± 0.06 | 3.59 ± 0.16 | 7.04 ± 0.31 | 14.33 ± 0.54 |
| | 4 | 1.86 ± 0.07 | 3.61 ± 0.16 | 7.00 ± 0.35 | 14.43 ± 0.56 |
| | 5 | 1.96 ± 0.07 | 3.99 ± 0.15 | 7.88 ± 0.34 | 15.71 ± 0.54 |
| | 6 | 1.84 ± 0.08 | 3.77 ± 0.14 | 7.61 ± 0.29 | 15.41 ± 0.55 |
| | 7 | 1.83 ± 0.10 | 3.72 ± 0.16 | 7.53 ± 0.32 | 15.27 ± 0.60 |
| | 8 | 1.80 ± 0.09 | 3.72 ± 0.19 | 7.65 ± 0.29 | 15.38 ± 0.56 |
| | 9 | 1.83 ± 0.07 | 3.61 ± 0.24 | 6.93 ± 0.48 | 14.20 ± 0.71 |
| | 10 | 1.93 ± 0.07 | 3.89 ± 0.15 | 7.85 ± 0.23 | 16.00 ± 0.00 |
| | 11 | 1.94 ± 0.07 | 3.87 ± 0.16 | 7.83 ± 0.23 | 16.00 ± 0.00 |
| | 12 | 1.95 ± 0.08 | 3.78 ± 0.19 | 7.70 ± 0.35 | 16.00 ± 0.00 |
| | 13 | 1.95 ± 0.07 | 3.90 ± 0.17 | 7.89 ± 0.26 | 16.00 ± 0.00 |
| | 14 | 1.93 ± 0.07 | 3.85 ± 0.17 | 7.82 ± 0.26 | 16.00 ± 0.00 |
| | 15 | 1.96 ± 0.07 | 3.93 ± 0.14 | 7.88 ± 0.20 | 16.00 ± 0.00 |
| | 16 | 1.97 ± 0.07 | 3.91 ± 0.15 | 7.88 ± 0.25 | 16.00 ± 0.00 |
| | 17 | 1.97 ± 0.07 | 3.93 ± 0.14 | 7.90 ± 0.20 | 16.00 ± 0.00 |
| | 18 | 2.01 ± 0.06 | 3.97 ± 0.14 | 7.98 ± 0.18 | 16.00 ± 0.00 |
| | 19 | 1.83 ± 0.06 | 3.64 ± 0.13 | 7.32 ± 0.26 | 15.08 ± 0.46 |
| | 20 | 2.00 ± 0.00 | 4.00 ± 0.00 | 8.00 ± 0.00 | 16.00 ± 0.00 |
| | 21 | 2.00 ± 0.00 | 4.00 ± 0.00 | 8.00 ± 0.00 | 16.00 ± 0.00 |
| | 22 | 2.00 ± 0.00 | 4.00 ± 0.00 | 8.00 ± 0.00 | 16.00 ± 0.00 |

Table 3: **Layer-wise $p$-distribution of TinyImageNet-trained $L_p$-masks** All values (Median $\pm$ Stdev) are calculated with $p$ of all $L_p$-masks in each layer, from 5 different trials of TinyImageNet-trained models.

| | | TinyImageNet | | | |
|---|---|---|---|---|---|
| **Model** | **Layer** | $L_p$-**Conv (p=2)** | $L_p$-**Conv (p=4)** | $L_p$-**Conv (p=8)** | $L_p$-**Conv (p=16)** |
| AlexNet | 1 | 1.91 $\pm$ 0.22 | 3.62 $\pm$ 0.31 | 7.21 $\pm$ 0.41 | 14.57 $\pm$ 0.89 |
| | 2 | 1.99 $\pm$ 0.06 | 3.91 $\pm$ 0.28 | 7.52 $\pm$ 0.70 | 15.21 $\pm$ 1.33 |
| | 3 | 1.91 $\pm$ 0.11 | 3.46 $\pm$ 0.25 | 7.15 $\pm$ 0.66 | 14.78 $\pm$ 1.34 |
| | 4 | 1.71 $\pm$ 0.16 | 2.97 $\pm$ 0.37 | 6.51 $\pm$ 0.58 | 13.86 $\pm$ 0.91 |
| | 5 | 1.70 $\pm$ 0.18 | 2.82 $\pm$ 0.47 | 6.34 $\pm$ 0.75 | 13.41 $\pm$ 1.14 |
| ResNet-18 | 1 | 2.12 $\pm$ 0.33 | 4.04 $\pm$ 0.41 | 7.92 $\pm$ 0.70 | 15.93 $\pm$ 1.82 |
| | 2 | 2.07 $\pm$ 0.08 | 4.02 $\pm$ 0.17 | 7.83 $\pm$ 0.49 | 16.56 $\pm$ 1.32 |
| | 3 | 1.98 $\pm$ 0.08 | 3.75 $\pm$ 0.17 | 7.36 $\pm$ 0.39 | 16.97 $\pm$ 1.42 |
| | 4 | 1.99 $\pm$ 0.09 | 3.75 $\pm$ 0.17 | 7.35 $\pm$ 0.36 | 16.92 $\pm$ 1.29 |
| | 5 | 1.98 $\pm$ 0.08 | 3.68 $\pm$ 0.18 | 7.19 $\pm$ 0.30 | 17.19 $\pm$ 1.26 |
| | 6 | 2.02 $\pm$ 0.10 | 3.85 $\pm$ 0.18 | 7.45 $\pm$ 0.28 | 17.52 $\pm$ 1.29 |
| | 7 | 1.84 $\pm$ 0.11 | 3.35 $\pm$ 0.22 | 6.77 $\pm$ 0.42 | 15.33 $\pm$ 1.00 |
| | 8 | 1.90 $\pm$ 0.08 | 4.05 $\pm$ 0.23 | 8.83 $\pm$ 0.31 | 17.29 $\pm$ 0.32 |
| | 9 | 1.89 $\pm$ 0.12 | 3.48 $\pm$ 0.25 | 7.06 $\pm$ 0.43 | 15.96 $\pm$ 1.07 |
| | 10 | 1.98 $\pm$ 0.10 | 3.63 $\pm$ 0.24 | 7.17 $\pm$ 0.41 | 16.97 $\pm$ 1.20 |
| | 11 | 2.02 $\pm$ 0.11 | 3.82 $\pm$ 0.23 | 7.40 $\pm$ 0.37 | 17.15 $\pm$ 1.18 |
| | 12 | 1.82 $\pm$ 0.11 | 3.30 $\pm$ 0.23 | 6.71 $\pm$ 0.35 | 15.50 $\pm$ 1.02 |
| | 13 | 1.75 $\pm$ 0.06 | 3.77 $\pm$ 0.18 | 8.64 $\pm$ 0.25 | 17.16 $\pm$ 0.27 |
| | 14 | 1.84 $\pm$ 0.12 | 3.39 $\pm$ 0.23 | 7.07 $\pm$ 0.37 | 16.47 $\pm$ 1.08 |
| | 15 | 1.93 $\pm$ 0.12 | 3.57 $\pm$ 0.28 | 7.24 $\pm$ 0.42 | 17.77 $\pm$ 1.15 |
| | 16 | 2.01 $\pm$ 0.11 | 3.82 $\pm$ 0.24 | 7.56 $\pm$ 0.41 | 17.49 $\pm$ 1.10 |
| | 17 | 1.75 $\pm$ 0.08 | 3.19 $\pm$ 0.16 | 6.68 $\pm$ 0.26 | 15.60 $\pm$ 0.70 |
| | 18 | 1.82 $\pm$ 0.06 | 4.28 $\pm$ 0.17 | 9.25 $\pm$ 0.18 | 17.59 $\pm$ 0.16 |
| | 19 | 1.78 $\pm$ 0.10 | 3.25 $\pm$ 0.16 | 7.18 $\pm$ 0.38 | 17.55 $\pm$ 0.96 |
| | 20 | 1.69 $\pm$ 0.06 | 3.57 $\pm$ 0.13 | 8.29 $\pm$ 0.27 | 18.37 $\pm$ 0.67 |
| ResNet-34 | 1 | 2.11 $\pm$ 0.24 | 4.02 $\pm$ 0.36 | 7.92 $\pm$ 0.59 | 15.75 $\pm$ 1.97 |
| | 2 | 2.10 $\pm$ 0.08 | 4.04 $\pm$ 0.17 | 7.93 $\pm$ 0.50 | 16.90 $\pm$ 1.29 |
| | 3 | 2.04 $\pm$ 0.09 | 3.87 $\pm$ 0.20 | 7.48 $\pm$ 0.53 | 17.23 $\pm$ 1.30 |
| | 4 | 2.04 $\pm$ 0.09 | 3.88 $\pm$ 0.19 | 7.50 $\pm$ 0.49 | 17.14 $\pm$ 1.28 |
| | 5 | 2.02 $\pm$ 0.09 | 3.80 $\pm$ 0.22 | 7.36 $\pm$ 0.40 | 17.41 $\pm$ 1.27 |
| | 6 | 2.02 $\pm$ 0.08 | 3.80 $\pm$ 0.17 | 7.39 $\pm$ 0.35 | 17.48 $\pm$ 1.32 |
| | 7 | 2.05 $\pm$ 0.08 | 3.83 $\pm$ 0.19 | 7.35 $\pm$ 0.43 | 17.65 $\pm$ 1.18 |
| | 8 | 2.10 $\pm$ 0.10 | 3.96 $\pm$ 0.17 | 7.53 $\pm$ 0.32 | 18.15 $\pm$ 1.33 |
| | 9 | 1.87 $\pm$ 0.12 | 3.46 $\pm$ 0.25 | 6.93 $\pm$ 0.53 | 15.58 $\pm$ 0.98 |
| | 10 | 1.86 $\pm$ 0.07 | 3.93 $\pm$ 0.20 | 8.67 $\pm$ 0.30 | 17.14 $\pm$ 0.30 |
| | 11 | 1.87 $\pm$ 0.12 | 3.54 $\pm$ 0.24 | 7.21 $\pm$ 0.44 | 16.33 $\pm$ 1.07 |
| | 12 | 1.96 $\pm$ 0.13 | 3.67 $\pm$ 0.28 | 7.27 $\pm$ 0.48 | 17.16 $\pm$ 1.17 |
| | 13 | 1.96 $\pm$ 0.12 | 3.67 $\pm$ 0.22 | 7.34 $\pm$ 0.42 | 17.24 $\pm$ 1.21 |
| | 14 | 2.06 $\pm$ 0.11 | 3.88 $\pm$ 0.25 | 7.64 $\pm$ 0.55 | 17.96 $\pm$ 1.18 |
| | 15 | 1.98 $\pm$ 0.11 | 3.75 $\pm$ 0.22 | 7.48 $\pm$ 0.38 | 17.63 $\pm$ 1.25 |
| | 16 | 2.12 $\pm$ 0.12 | 4.05 $\pm$ 0.25 | 7.98 $\pm$ 0.61 | 18.33 $\pm$ 1.11 |
| | 17 | 2.11 $\pm$ 0.13 | 4.03 $\pm$ 0.24 | 7.62 $\pm$ 0.50 | 18.18 $\pm$ 1.24 |
| | 18 | 1.88 $\pm$ 0.12 | 3.47 $\pm$ 0.26 | 6.90 $\pm$ 0.42 | 15.96 $\pm$ 1.15 |
| | 19 | 1.73 $\pm$ 0.06 | 3.73 $\pm$ 0.18 | 8.61 $\pm$ 0.25 | 17.13 $\pm$ 0.26 |
| | 20 | 1.89 $\pm$ 0.12 | 3.52 $\pm$ 0.24 | 7.20 $\pm$ 0.39 | 16.64 $\pm$ 1.10 |
| | 21 | 1.94 $\pm$ 0.15 | 3.56 $\pm$ 0.32 | 7.27 $\pm$ 0.51 | 17.63 $\pm$ 1.20 |
| | 22 | 1.90 $\pm$ 0.11 | 3.61 $\pm$ 0.21 | 7.40 $\pm$ 0.38 | 17.34 $\pm$ 1.13 |
| | 23 | 1.99 $\pm$ 0.12 | 3.81 $\pm$ 0.27 | 7.55 $\pm$ 0.53 | 18.43 $\pm$ 1.07 |
| | 24 | 1.87 $\pm$ 0.11 | 3.64 $\pm$ 0.19 | 7.57 $\pm$ 0.34 | 17.72 $\pm$ 1.07 |
| | 25 | 2.00 $\pm$ 0.10 | 3.92 $\pm$ 0.22 | 7.91 $\pm$ 0.47 | 18.69 $\pm$ 0.85 |
| | 26 | 1.85 $\pm$ 0.09 | 3.71 $\pm$ 0.16 | 7.69 $\pm$ 0.29 | 17.94 $\pm$ 0.95 |
| | 27 | 2.03 $\pm$ 0.09 | 4.02 $\pm$ 0.20 | 8.18 $\pm$ 0.48 | 18.64 $\pm$ 0.82 |
| | 28 | 1.86 $\pm$ 0.08 | 3.77 $\pm$ 0.16 | 7.79 $\pm$ 0.27 | 17.98 $\pm$ 0.88 |
| | 29 | 2.07 $\pm$ 0.10 | 4.11 $\pm$ 0.21 | 8.35 $\pm$ 0.53 | 18.54 $\pm$ 0.83 |
| | 30 | 1.95 $\pm$ 0.09 | 3.90 $\pm$ 0.20 | 7.69 $\pm$ 0.38 | 18.75 $\pm$ 1.07 |
| | 31 | 1.98 $\pm$ 0.10 | 3.71 $\pm$ 0.21 | 7.31 $\pm$ 0.29 | 15.82 $\pm$ 0.70 |
| | 32 | 1.86 $\pm$ 0.07 | 4.41 $\pm$ 0.22 | 9.44 $\pm$ 0.21 | 17.74 $\pm$ 0.18 |
| | 33 | 1.82 $\pm$ 0.09 | 3.31 $\pm$ 0.16 | 6.90 $\pm$ 0.30 | 16.64 $\pm$ 1.02 |
| | 34 | 1.88 $\pm$ 0.09 | 3.55 $\pm$ 0.19 | 7.43 $\pm$ 0.29 | 18.29 $\pm$ 1.02 |
| | 35 | 1.87 $\pm$ 0.12 | 3.47 $\pm$ 0.19 | 7.54 $\pm$ 0.41 | 17.82 $\pm$ 0.97 |
| | 36 | 1.90 $\pm$ 0.06 | 3.78 $\pm$ 0.13 | 8.07 $\pm$ 0.26 | 18.92 $\pm$ 0.69 |
| VGG-16 | 1 | 2.07 $\pm$ 0.16 | 4.11 $\pm$ 0.24 | 8.29 $\pm$ 0.72 | 17.52 $\pm$ 1.71 |
| | 2 | 2.15 $\pm$ 0.10 | 4.17 $\pm$ 0.21 | 8.18 $\pm$ 0.52 | 17.27 $\pm$ 1.11 |
| | 3 | 2.08 $\pm$ 0.11 | 4.09 $\pm$ 0.23 | 7.99 $\pm$ 0.54 | 17.38 $\pm$ 1.30 |
| | 4 | 2.15 $\pm$ 0.10 | 4.12 $\pm$ 0.21 | 7.92 $\pm$ 0.49 | 17.76 $\pm$ 1.16 |
| | 5 | 1.95 $\pm$ 0.10 | 3.79 $\pm$ 0.22 | 7.65 $\pm$ 0.52 | 17.52 $\pm$ 1.39 |
| | 6 | 1.98 $\pm$ 0.12 | 3.78 $\pm$ 0.29 | 7.55 $\pm$ 0.56 | 18.26 $\pm$ 1.25 |
| | 7 | 2.04 $\pm$ 0.14 | 3.98 $\pm$ 0.31 | 7.72 $\pm$ 0.60 | 18.05 $\pm$ 1.36 |
| | 8 | 1.95 $\pm$ 0.13 | 3.72 $\pm$ 0.30 | 7.50 $\pm$ 0.54 | 16.84 $\pm$ 1.32 |
| | 9 | 1.91 $\pm$ 0.14 | 3.68 $\pm$ 0.29 | 7.69 $\pm$ 0.48 | 18.04 $\pm$ 1.27 |
| | 10 | 2.00 $\pm$ 0.15 | 3.90 $\pm$ 0.29 | 7.91 $\pm$ 0.50 | 18.38 $\pm$ 1.64 |
| | 11 | 1.82 $\pm$ 0.11 | 3.43 $\pm$ 0.19 | 7.28 $\pm$ 0.36 | 16.61 $\pm$ 1.28 |
| | 12 | 1.83 $\pm$ 0.10 | 3.51 $\pm$ 0.18 | 7.52 $\pm$ 0.27 | 16.78 $\pm$ 1.29 |
| | 13 | 1.96 $\pm$ 0.07 | 3.86 $\pm$ 0.14 | 7.64 $\pm$ 0.26 | 16.09 $\pm$ 0.80 |
| ConvNeXt-T | 1 | 1.98 $\pm$ 0.20 | 4.01 $\pm$ 0.37 | 8.16 $\pm$ 1.06 | 16.22 $\pm$ 2.12 |
| | 2 | 1.95 $\pm$ 0.10 | 3.70 $\pm$ 0.22 | 7.42 $\pm$ 0.49 | 14.91 $\pm$ 0.92 |
| | 3 | 1.97 $\pm$ 0.11 | 3.71 $\pm$ 0.21 | 7.26 $\pm$ 0.45 | 14.63 $\pm$ 0.93 |
| | 4 | 1.96 $\pm$ 0.12 | 3.68 $\pm$ 0.23 | 7.25 $\pm$ 0.45 | 15.15 $\pm$ 0.99 |
| | 5 | 1.88 $\pm$ 0.15 | 3.92 $\pm$ 0.36 | 7.66 $\pm$ 1.09 | 15.48 $\pm$ 2.55 |
| | 6 | 1.94 $\pm$ 0.11 | 3.65 $\pm$ 0.22 | 7.11 $\pm$ 0.44 | 14.38 $\pm$ 1.01 |
| | 7 | 1.96 $\pm$ 0.09 | 3.58 $\pm$ 0.22 | 7.01 $\pm$ 0.48 | 13.86 $\pm$ 0.88 |
| | 8 | 1.95 $\pm$ 0.11 | 3.59 $\pm$ 0.24 | 7.02 $\pm$ 0.45 | 14.12 $\pm$ 0.96 |
| | 9 | 1.83 $\pm$ 0.13 | 3.71 $\pm$ 0.47 | 7.11 $\pm$ 1.02 | 14.22 $\pm$ 2.32 |
| | 10 | 1.94 $\pm$ 0.09 | 3.51 $\pm$ 0.20 | 6.64 $\pm$ 0.43 | 13.76 $\pm$ 0.89 |
| | 11 | 1.93 $\pm$ 0.09 | 3.50 $\pm$ 0.20 | 6.68 $\pm$ 0.42 | 13.87 $\pm$ 0.84 |
| | 12 | 1.93 $\pm$ 0.10 | 3.50 $\pm$ 0.20 | 6.62 $\pm$ 0.40 | 13.56 $\pm$ 0.84 |
| | 13 | 1.94 $\pm$ 0.11 | 3.54 $\pm$ 0.20 | 6.64 $\pm$ 0.39 | 13.75 $\pm$ 0.80 |
| | 14 | 1.94 $\pm$ 0.10 | 3.53 $\pm$ 0.21 | 6.66 $\pm$ 0.40 | 13.68 $\pm$ 0.77 |
| | 15 | 1.93 $\pm$ 0.11 | 3.56 $\pm$ 0.22 | 6.71 $\pm$ 0.40 | 13.75 $\pm$ 0.75 |
| | 16 | 1.94 $\pm$ 0.11 | 3.58 $\pm$ 0.19 | 6.80 $\pm$ 0.40 | 14.14 $\pm$ 0.85 |
| | 17 | 1.94 $\pm$ 0.11 | 3.59 $\pm$ 0.29 | 7.03 $\pm$ 0.42 | 14.44 $\pm$ 0.88 |
| | 18 | 1.95 $\pm$ 0.10 | 3.61 $\pm$ 0.25 | 7.01 $\pm$ 0.39 | 14.55 $\pm$ 0.80 |
| | 19 | 1.74 $\pm$ 0.08 | 3.64 $\pm$ 0.25 | 7.32 $\pm$ 0.51 | 15.15 $\pm$ 1.17 |
| | 20 | 1.87 $\pm$ 0.10 | 3.62 $\pm$ 0.21 | 7.20 $\pm$ 0.34 | 14.69 $\pm$ 0.77 |
| | 21 | 1.87 $\pm$ 0.10 | 3.65 $\pm$ 0.19 | 7.21 $\pm$ 0.35 | 14.55 $\pm$ 0.77 |
| | 22 | 1.92 $\pm$ 0.09 | 3.77 $\pm$ 0.17 | 7.45 $\pm$ 0.33 | 15.02 $\pm$ 0.76 |

## A.18 LEARNED $L_p$-MASK PROPERTIES ($p$=16)

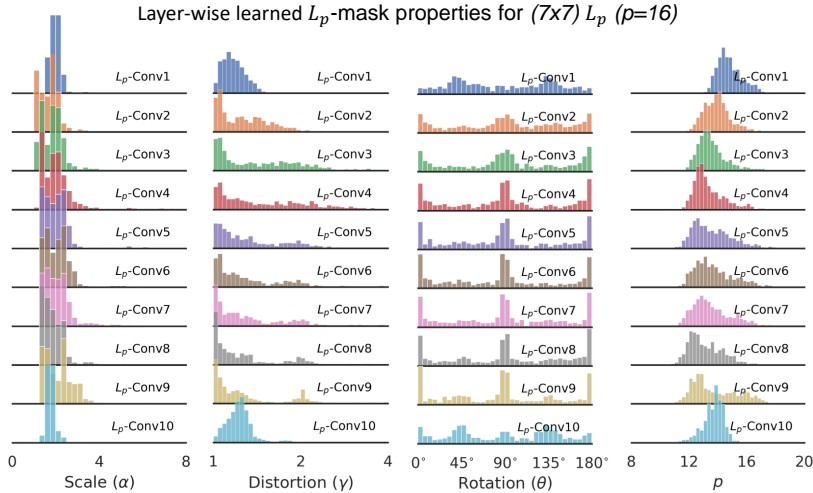

Figure 16: **Layer-wise distribution of learned $L_p$-mask properties ($p$=16)**

## A.19 STATISTICS FOR COMPARISON OF OPTIMIZED MPND $p$ VALUES IN ARTIFICIAL AND BIOLOGICAL RFs

Table 4: Holm-Bonferroni corrected multiple comparisons of each RFs using Welch's t-test. Lower diagonal elements denote corrected statistical p-values and upper diagonal elements denote degree of significance (n.s, not significant; *, 0.01<p<0.05; **, 0.001<p<0.01; ***, p<0.001; )

| Welch's t-test (corrected p-values) | Mouse V1 Layer 2/3 | Trained + Image | Trained + Noise | Untrained + Noise | Untrained + Image |
|---|---|---|---|---|---|
| **Mouse V1 Layer 2/3** | - | * | * | *** | *** |
| **Trained + Image** | 1.10e-2 | - | n.s | *** | *** |
| **Trained + Noise** | 1.06e-2 | 7.6e-1 | - | *** | *** |
| **Untrained + Noise** | 1.23e-15 | 4.13e-28 | 1.49e-25 | - | *** |
| **Untrained + Image** | 8.94e-14 | 1.59e-25 | 6.73e-23 | 1.39e-22 | - |

## A.20 IMPACT OF $L_p$-CONVOLUTION (P=2) ON OTHER CNN ARCHITECTURES

Table 5: Top-1 performance (mean±std, 5 trials) on the CIFAR-100 datasets with $L_p$-Convolution applied in ConvNeXt-V2-T (Woo et al., 2023), ResNet-50 (He et al., 2016), ResNext-50 (Xie et al., 2017) and DenseNet-121 (Huang et al., 2017). The symbol ✓ indicates $L_p$-Converted ($p_{\text{init}} = 2$) or not. '***' denotes statistical comparison using Welch's t-test ($p < 0.001$).

| $L_p$-Conv | ConvNeXt-V2-T | CIFAR-100 ResNet-50 | ResNeXt-50 | DenseNet-121 |
|---|---|---|---|---|
| - | 64.26 ± 0.41 | 73.17 ± 0.23 | 73.55 ± 0.57 | 74.12 ± 0.16 |
| ✓ | *** 65.58 ± 0.25 | *** 76.66 ± 0.19 | *** 77.38 ± 0.36 | *** 77.14 ± 0.18 |

