# OpenReview forum: "Brain-inspired $L_p$-Convolution benefits large kernels and aligns better with visual cortex"
_ICLR.cc/2024/Conference — Submitted to ICLR 2024_

### Official Review · Reviewer_qr9c · 2023-10-20

**Soundness:** 4 excellent
**Presentation:** 3 good
**Contribution:** 2 fair
**Rating:** 5
**Confidence:** 4

**Summary:**

Inspired by the observation that the spatial distribution synaptic inputs in the early visual system of the brain is approximately Gaussian, the authors propose a masking strategy for convolution filters in convnets for image recognition. They observe that such (approximately) Gaussian masking enables convnets to learn with larger filters, leading to improved performance on the Sudoku challenge and a number of small-scale image classification tasks. They also show that networks trained with Gaussian masked filters exhibit slightly increased representational similarity to the mouse visual system.

**Strengths:**

+ Creative inductive bias transfer from biology to machine learning
 + Overall well-written paper
 + Well-motivated and well-executed experiments

**Weaknesses:**

1. Effect sizes are quite small
 1. A number of details on the experiments remain unclear even after screening the appendix

**Questions:**

I am somewhat torn on the paper. While I appreciate the clear motivation and hypothesis, I am somewhat underwhelmed by the results and/or how they are (over?)sold in the paper. If the authors can provide convincing answers to the following questions, I am willing to adjust my score.

### 1. Effect size

While I very much appreciate the three main experiments presented in Fig. 3, Table 1 and Fig. 5, I am somewhat underwhelmed by the effect sizes.

 a) In the Sudoku challenge, the main argument seems to be about the imbalance between row/column and block accuracy, but the difference in accuracy is <10% in all cases, which doesn't strike me as a particularly worrisome imbalance. In particular, the difference between p=2 and 7x7 (Large) is <1% (although I'm not sure what the latter model exactly is; see below). Could this be a matter of presentation and the differences would be much clearer if you looked at error rates instead of accuracy? Could you explain why you think these results are important given such small effect size?

 b) In the image classification experiments, there is a clearly significant improvement due to Lp-Conv, but not on all architectures and again the improvement is small (a few percent). Given that architectural modifications alone can now push accuracy on CIFAR-100 >90% (https://arxiv.org/abs/2304.05350v2), the 2–3% improvements in the 60–70% range feel a bit insignificant. Can you explain why you think your approach is still worthwhile? Would we expect similar gains if we included the Gaussian masking inductive bias into more modern architectures?

 c) In the representational similarity experiments the improvements due to p=2 are in the range of 0.5–2%. Again, why are such small differences relevant from a scientific point of view if the differences between, e.g., AlexNet and ConvNeXt-T are of the same order of magnitude?


### Experimental details

 a) Are the masks applied after training to the trained weights or during training? If the latter, does it change anything about what the network learns? It could just learn larger weights W where the mask is small, leading to the exact same network as without masking. Why is this not happening?

 b) It is not clear to me what the different networks in Fig. 3 are. It's clear that 3x3 and 7x7 refer to the kernel size. But what does "Large" refer to and what is the Lp^{\dagger} model?

 c) As far as I understand Fig. 3, p=2 refers to p being initialized as 2 but then optimized as a trainable parameter, correct? If so, what was p at the end of training for the three models with p={2, 16, 256}? This question actually applies to all experiments.

 d) Table 1: ResNet has larger kernels than 3x3 in the first layer. Are they kept the same in all variants and only the 3x3 kernels in the later layers are modified?

 e) Fig. 5: I did not fully grasp what "Max. SSM" exactly refers to. In panel a) five brain areas are colored; in panel b) five network architectures. Presumably you compared all architectures against all brain areas. What exactly is being reported here? Why is there not one such plot as in b) per brain area?


### Other questions

 a) Why is the spatial distribution of synapses the right thing to compare to weights in a CNN? The presynaptic neurons also have spatially extended receptive fields (RFs), which means the V1 neuron's RF envelope in pixel space is not the same as the retinotopically mapped synaptic locations.

---

> ### Author Response · Authors · 2023-11-20
>
> We appreciate the valuable feedback from the reviewer. Following are pointwise responses to the reviewer.
>
> ---
> > __Re.1)__ In the Sudoku challenge, the main argument seems to be about the imbalance between row/column and block accuracy, but the difference in accuracy is <10% in all cases, which doesn't strike me as a particularly worrisome imbalance.  In particular, the difference between p=2 and 7x7 (Large) is <1%. Could this be a matter of presentation and the differences would be much clearer if you looked at error rates instead of accuracy? Could you explain why you think these results are important given such a small effect size?
>
> Thank you for the reviewer's critical point. We understand the reviewer's point on the accuracy gap ($\sim$10%; row/column vs square) may not seem worrisome with respect to the imbalance itself. However, this is slightly different from our intention. We would like to clarify our main claim.
>
> Our main claim here is that ___such a small imbalance can have a significant impact on a greater problem___. Considering the fact that a Sudoku puzzle is solved only when all 81 numbers align correctly, a single error in any of the 18 rows/columns ($\approx$6% error) results in a Sudoku unsolved. We believe our experimental result in __Fig. 3b__ supports this point; small imbalances in row/col and square accuracies (1$\sim$10%) can have a significant impact on overall Sudoku accuracy (20$\sim$60%).
>
> To convey this point more clearly, we have revised our manuscript by changing the figure, and subsection titles in Section 4 as follows.
> - (Fig. 3 title) $L_p$-convolution enhances Sudoku solving efficiency by effectively balancing accuracy between square and row-column puzzles
> - (subsection 3 title) $L_p$-convolution in Sudoku solving: balancing square and row-column imbalances
>
> ---
> > __Re.2)__ In the image classification experiments, there is a clearly significant improvement due to Lp-Conv, but not on all architectures and again the improvement is small. Given that architectural modifications alone can now push accuracy on CIFAR-100 >90%, the 2–3% improvements in the 60–70% range feel a bit insignificant. Can you explain why you think your approach is still worthwhile? Would we expect similar gains if we included the Gaussian masking inductive bias into more modern architectures?
>
> Thank you for sharing the impressive paper. While architectural modifications alone can achieve remarkable performance gains, these results can be seen as the outcome of a combination of various excellent methods. ___Our work can be considered as the discovery of a new method that can be combined with such architectural modifications.___ From this perspective, we believe our work is sufficiently valuable.
>
> In response to the reviewer's question we applied $L_p$-Conv (p=2) on the ConvNeXt-V2 (Woo et al., CVPR 2023) and other higher accuracy range (70~80%) models with CIFAR100 as follows (See __Appendix A.20, Table 5__):
>
> |$L_p$-Conv|ConvNeXt-V2-T|ResNet-50|ResNeXt-50|DenseNet-121|
> |:-:|:-:|:-:|:-:|:-:|
> |-|64.26±0.41|73.17±0.23|73.55±0.57|74.12±0.16|
> |$\checkmark$|65.58±0.25|76.66±0.19|77.38±0.36|77.14±0.18|
>
> Our method enhances performance in 1) modern CNN architecture as well as 2) higher accuracy models with 3~4% improvement. These results highlight the versatility of our $L_p$-convolution method to a diverse range of architectures. We believe that integrating our Gaussian masking inductive bias into various architectures could yield similar gains, further affirming the importance of methodological innovations alongside architectural advancements.
>
> ---
> > __Re.3)__ In the representational similarity experiments the improvements due to p=2 are in the range of 0.5–2%. Again, why are such small differences relevant from a scientific point of view if the differences between, e.g., AlexNet and ConvNeXt-T are of the same order of magnitude?
>
> Thank you for your thoughtful question. In the field of computational neuroscience, the use of artificial systems as a comparative tool has proven invaluable, particularly in understanding the brain's visual processing [1, 2]. Although AlexNet and ConvNeXt-T show similar levels of improvement, the reasons behind these gains are quite intricate, especially when considering structural differences. Our research focuses on unraveling these complexities. Specifically, the consistent improvement pattern across various models with the adjustment of the $p$ value is compelling. ___It suggests a shared underlying mechanism in visual information processing___, potentially mirroring the Gaussian-like receptive fields observed in the human brain. I hope this answers your question!
>
> [1] Kriegeskorte et al., "Representational similarity analysis - connecting the branches of systems neuroscience," Frontiers in Systems Neuroscience, 2008.
> [2] Cadieu et al., "Deep neural networks rival the representation of primate IT cortex for core visual object recognition," PLoS Computational Biology, 2014.

---

> ### Author Response · Authors · 2023-11-20
> **Official Comment by Authors - 2**
>
> > __Re.4)__ Experimental details a) Are the masks applied after training to the trained weights or during training? If the latter, does it change anything about what the network learns? It could just learn larger weights W where the mask is small, leading to the exact same network as without masking. Why is this not happening?
>
> In response to the reviewer's question, we clarify that the $ L_p$-mask is trained with weight simultaneously. With proper initialization of $L_p$-mask, ___it imposes a spatial constraint on the weights which leads to a change in learning dynamics___. Since both mask and weights are trained with standard backpropagation after the forward process ($Y = \phi(X*(W \odot M))$, Eqn. 4), $L_p$-mask effectively scales down the influence of peripheral weights.  This setup leads to a distinct learning behavior compared to a scenario without a mask, as the mask evolves (Fig. 2, 4, and 8) together with weight update.
>
> ---
> > __Re.5)__ Experimental details b) It is not clear to me what the different networks in Fig. 3 are. It's clear that 3x3 and 7x7 refer to the kernel size. But what does "Large" refer to and what is the Lp^{\dagger} model?
>
> Apologies for the insufficient information. We have revised the manuscript's Fig. 3b to address this.
> - (new in Fig. 3b caption) '(3x3)' or '(7x7)' denotes the size kernel. '$L_p^\dagger$' denotes parameters of $L_p$-mask is frozen. 'Large' denotes a simple enlargement of the kernel, without a mask.
>
> ---
> > __Re.6)__ Experimental details c) As far as I understand Fig. 3, p=2 refers to p being initialized as 2 but then optimized as a trainable parameter, correct? If so, what was p at the end of training for the three models with p={2, 16, 256}? This question actually applies to all experiments.
>
> Thank you for your question regarding the optimization of $p$.
> - In the case of $L_p^\dagger (p=256)$, the parameters are fixed, meaning that $p$ remains constant at 256 throughout the training process.
> - For the $p=16$ scenario, we have detailed the properties of $L_p$-masks in the revised manuscript, specifically in __Appendix A.18__.
> - Additionally, the distribution of $p$ values across all experiments is comprehensively visualized in __Appendix A.17__ of the revised manuscript.
>
> ---
> > __Re.7)__ Experimental details d) Table 1: ResNet has larger kernels than 3x3 in the first layer. Are they kept the same in all variants and only the 3x3 kernels in the later layers are modified?
>
> All kernels are ___equally modified___.
> - In TinyImageNet, ResNet has a 7x7 kernel size in the first layer and is equally enlarged as 15x15 in '(Large)' or '$L_p$-Conv'.
> - In CIFAR-100 due to its small image size (32 x 32), ResNet has a 3x3 kernel size in the first layer [1, 2].  This is enlarged as 7x7 in '(Large)' or '$L_p$-Conv'.
> [1] He, Kaiming, et al. "Deep residual learning for image recognition." CVPR. 2016.
> [2] https://github.com/weiaicunzai/pytorch-cifar100/blob/master/models/resnet.py
> ---
>
> > __Re.8)__ Experimental details e) Fig. 5: I did not fully grasp what "Max. SSM" exactly refers to. In panel a) five brain areas are colored; in panel b) five network architectures. Presumably you compared all architectures against all brain areas. What exactly is being reported here? Why is there not one such plot as in b) per brain area?
>
> "Max. SSM" refers to the ___maximum SSM value obtained from our pairwise comparisons___ of all network architectures against each brain area, following the methodology established by Bakhtiari et al. in 2021. As you correctly interpreted, these comprehensive comparisons are detailed in __Appendix A.12, Figure 10__ of our manuscript. Due to limited space, we included this analysis in the Appendix to provide a clear presentation of our result.
>
> - Bakhtiari, Shahab, et al. "The functional specialization of visual cortex emerges from training parallel pathways with self-supervised predictive learning." Advances in Neural Information Processing Systems 34 (2021)
>
> ---
> > __Re.9)__ Other questions a) Why is the spatial distribution of synapses the right thing to compare to weights in a CNN? The presynaptic neurons also have spatially extended receptive fields (RFs), which means the V1 neuron's RF envelope in pixel space is not the same as the retinotopically mapped synaptic locations.applies to all experiments.
>
> Thank you for your insightful feedback. We understand the limitations in our approach, particularly that the spatial distribution of V1 synapses might not perfectly mirror the weights in a CNN. The idea of comparing retinotopically mapped synaptic locations is an intriguing alternative. Our current method, aligning V1 synaptic patterns with CNN weight distributions, is an ___exploratory effort aimed at fostering better dialogue between the realms of biological and artificial vision___. Though it may not be a direct equivalence, this approach marks a significant step in our journey to understand the intricate parallels between these systems.

---

> ### Author Response · Authors · 2023-11-22
> **A gentle reminder**
>
> Dear Reviewer,
>
> Thank you for your insightful feedback on our manuscript. We have thoroughly revised it, emphasizing the clarification of effect sizes and experimental details as per your suggestions. We hope these revisions meet your expectations, and we would greatly value any additional comments or thoughts you might have.

---

### Official Review · Reviewer_Bf1L · 2023-10-31

**Soundness:** 3 good
**Presentation:** 4 excellent
**Contribution:** 3 good
**Rating:** 8
**Confidence:** 3

**Summary:**

The authors propose adding a gaussian mask to the square kernels of CNNs/transformers, parametrized by the L_p metric, to give ANN receptive fields a flexibility similar to Biological NNs (BNNs). They find that these including these Lp masks improve network accuracy

**Strengths:**

The basic idea is very compelling.

The connection to BNNs is well supported.

The paper is very well-written.

The results table has +/- std devs.

**Weaknesses:**

I am left wondering if CNNs do this RF masking already, by different means.

I am unclear how the covariant matrix C is trainable. what is the update process?

Reviewer limitation: I am not well-versed in this literature, so I am assessing the paper itself with limited background.


Notes to ICLR:

1. Please include line numbers in the template. They make reviewing much easier!

2. Please reformat the default bibliography style to make searching the bib easier! eg numbered, last name first, initials only except for last name.

**Questions:**

General review context: Please note that I am simply another researcher, with some overlap of expertise with the content of the paper. In some cases my comments may reflect points that I believe are incorrect or incomplete. In most cases, my comments reflect spots where an average well-intentioned reader might stumble for various reasons, reducing the potential impact of the paper. The issue may be the text, or my finite understanding, or a combination. These comments point to opportunities to clarify and smooth the text to better convey the intended story. I urge the authors to decide how or whether to address these comments. I regret that the tone can come out negative even for a paper I admire; it's a time-saving mechanism for which I apologize.

Key comments:

Note: Addressing the two issues in "Weaknesses" above would be great. The rest of the comments can be handled as the authors see fit.

Bibliography: Perhaps reformat the bibliography for easier searching, eg numbered, last name first, initials only except for last name.

Abstract "gaussian-like structured sparsity": I don't think of gaussian weightings as "sparse", since few values get sent to zero. Is this the correct technical term for what is happening?

1. paragraph 1, "LecCun and ... by introducing backprop": Backprop was not LeCun. I think more complete citations are needed here.

1. paragraph 1, "alexNet": CNNs blew up with the convergence of backprop, CNNs, big data sets, and GPUs that enabled training them.

1. Paragraph 2, list of similarities: A 4th important similarity is local RFs (as you note in the next paragraph).

Provided code :)

2. Multivariate p-general... "the reference position of the RF": Do you mean "reference position of a pixel in the RF?". If not, I am confused.

Eqn 1: Is the superscript "p" standard notation? I think I usually see L_p as ||   ||_p (subscript only).

Covariate matrix C: does this raise dimensionality issues, since C includes d^2 new parameters per filter? (doubles the number of free parameters I think).

Fig 1 e: this shows up too small on a printed page.

Fig 1 f: Did you try p < 2? It looks like the masks converge to rectangles (all weights in mask the same) at low p, so that the difference between 4 and 16 is slight.

3. "introduce L_p convolution": clever mechanism.

Just after eqn 4 "C and p are trainable": I do not see how to do this. Is it explained in the paper?

"mask take one": I stumbled on this. Clearer might be "equal one", or "have the value one". Also, maybe note that this is the limit as p increases, and in fact is roughly attained at p = (some value).

4. Sudoku: Perhaps explain why this is a good test case for the method (since it is not usual, and the usual image examples come later).

Fig 3 b: the line colors/labels are unclear in these subplots. Perhaps make them bigger, or use dashed lines for some cases, or change to brighter colors.

Fig 4: What is the definition of "distortion"?

Fig 4 a: The different labels (alpha, theta) on the two sides of the mosaic are hard to decipher, especially on a printed page. Perhaps make the mosaic horizontal, or make labels larger, or break up the sets of 9 boxes with bolder lines, or provide clearer guidance in the caption.

Fig 4 a: what role do diagonal masks play in the Sudoku networks, given the problem's horizontal-vertical-box structure?

Fig 4 a: A crucial question for me: These RFs look a lot like what we often see in CNN papers. Is the L_p method generating something novel, or is it a new way of getting to what already happens?

Comparisons with base models:  It appears that adding an L_p mask at each filter effectively doubles the free parameter count while holding the filter size fixed. Might this account for the difference in accuracy scores?

Section 7: This looks like it should appear earlier in the paper - it is background.

---

> ### Author Response · Authors · 2023-11-20
>
> We are extremely happy with the reviewer's recognition of our contribution!
> Following are our point-wise responses to the reviewer.
>
> ---
> > __Re.1)__ I am left wondering if CNNs do this RF masking already, by different means.
>
> Thank you for your insightful query regarding receptive fields in CNNs.
>
> You are correct that many CNN studies discuss RFs, similar to what we've illustrated in our revised manuscript, specifically in __Appendix A.14, Figure 11a__. In traditional CNN models, these RFs are shaped by the network's trainable parameters – the kernel weights – and evolve naturally during the training process.
>
> However, our work takes a distinct approach. ___We've integrated structured sparsity constraints directly into the network's architecture___. This method doesn't just allow the RFs to emerge; it actively shapes them. This targeted shaping is crucial because it ensures the development of receptive fields that adhere to a specific pattern of structured sparsity. This approach is more than a byproduct of training; it's a strategic design choice.
>
> We believe that this novel method of guiding RF development represents a significant advancement in the field. It opens new avenues for understanding and utilizing receptive fields in CNN architectures, moving beyond traditional methods. Our hope is that this contribution will offer valuable insights into the potential of structured sparsity in neural networks.
>
> ---
> > __Re.2)__ I am unclear how the covariant matrix C is trainable. what is the update process?
>
> In our implementation, we set $C$ and $p$ as trainable parameters within the LpConv2d class, leveraging PyTorch's nn.Parameter (Pseudocod in __Appendix A.13__). These parameters are updated through the standard backpropagation process, just like other weights in the model. The mask in the convolution operation is dynamically generated, utilizing the current values of $C$ and $2$. Thanks to PyTorch's automatic gradient management, efficient training is ensured without the need for a specialized update mechanism.
>
> In the revised manuscript, we added footnote 3 on page 5 as follows.
> - (added) $C$ and $p$ are updated with the standard backpropagation process. $L_p$-mask is dynamically generated during forward process using $C$ and $p$.
>
> ---
> > __Re.3)__ Reviewer limitation: I am not well-versed in this literature, so I am assessing the paper itself with limited background.
> Notes to ICLR:
> Please include line numbers in the template. They make reviewing much easier!
> Please reformat the default bibliography style to make searching the bib easier! eg numbered, last name first, initials only except for last name.
>
> Even in such challenging circumstances, we express our sincere gratitude for the valuable feedback you have provided to us :)
>
> ---
> > __Re.4)__ General review context: Please note that I am simply another researcher, with some overlap of expertise with the content of the paper. In some cases my comments may reflect points that I believe are incorrect or incomplete. In most cases, my comments reflect spots where an average well-intentioned reader might stumble for various reasons, reducing the potential impact of the paper. The issue may be the text, or my finite understanding, or a combination. These comments point to opportunities to clarify and smooth the text to better convey the intended story. I urge the authors to decide how or whether to address these comments. I regret that the tone can come out negative even for a paper I admire; it's a time-saving mechanism for which I apologize.
>
> We were able to improve the quality of our manuscript with your thorough review, comment, and suggestion. We appreciate your contribution
>
> ---
> > __Re.5)__ Abstract "gaussian-like structured sparsity": I don't think of gaussian weightings as "sparse", since few values get sent to zero. Is this the correct technical term for what is happening?
>
> Our $L_p$ convolution in Eq. 4 adopts $p$-generalized normal distribution in Eq. 3. In essence, $L_p$ convolution introduces structured sparsity for sufficiently large values of $p$, as shown in Fig. 1. We agree that “gaussian-like structured sparsity” might lead to misunderstanding. Therefore, we revise the expression to “structured sparsity inspired by $p$-generalized normal distribution”.
>
> ---
> > __Re.6)__ paragraph 1, "LeCun and ... by introducing backprop": Backprop was not LeCun. I think more complete citations are needed here.
>
> We apologize for the misleading sentence. We do aware backdrop was not invented by LeCun.
>
> To clarify this, we revised the sentence as follows.
>
>
> - (before) LeCun and his colleagues built on this by introducing backpropagation, which led to the first implementation of a CNN (LeCun et al., 1989).
>
> - (after) Building upon the backpropagation (Werbos, 1974; Rumelhart et al., 1986), and the concept of weight sharing, LeCun and his colleagues applied it to CNNs, leading to the development of the first practical CNN, known as LeNet (LeCun et al., 1989).

---

> ### Author Response · Authors · 2023-11-20
> **Official Comment by Authors - 2**
>
> > __Re.7)__ paragraph 1, "alexNet": CNNs blew up with the convergence of backprop, CNNs, big data sets, and GPUs that enabled training them.
>
> We are glad that we could incorporate historical background information as well, thanks to your input.
>
> (before) The turning point for CNNs came in 2012 with the introduction of AlexNet, which outperformed existing models in the ImageNet challenge and brought CNNs into the spotlight (Krizhevsky et al., 2012).
>
> (after) Fueled by advancements in backpropagation, large datasets, and GPU training, the landmark success of AlexNet in 2012, which significantly outperformed existing models in the ImageNet challenge, marked a turning point for CNNs (Krizhevsky et al., 2012).
>
> ---
> > __Re.8)__ Paragraph 2, list of similarities: A 4th important similarity is local RFs (as you note in the next paragraph).
>
> Great point, we have listed your suggestion!
>
> (before) 1) RGB channels emulate retinal computations, 2) stacked layers correspond to various visual areas (such as V1, V2, V4, IT), and 3) deeper layers excel at detecting complex features
>
> (after) 1) RGB channels emulate retinal computations, 2) stacked layers correspond to various visual areas (such as V1, V2, V4, IT), 3) deeper layers excel at detecting complex features, and 4) local receptive fields.
>
> ---
> > __Re.9)__ Multivariate p-general... "the reference position of the RF": Do you mean "reference position of a pixel in the RF?". If not, I am confused.
>
> In our context, the reference position, represented by $\mathbf{s}$, is a specific point within this receptive field, with $ s_0 $ as its center. The term $ \Delta \mathbf{s} = \mathbf{s} - s_0 $ measures deviation from this central point. We use this framework to define the receptive field in our model, which is essential for introducing the MPND concept. We'll ensure to clarify this point further in our revised manuscript as following.
>
> (before) Let $\mathbf{s}$ be the $d$-dimensional random vectors that represent the reference position of the RF.
>
> (after) Let $\mathbf{s}$ represent the $d$-dimensional random vectors indicating specific points within RF.
>
> ---
> > __Re.10)__ Eqn 1: Is the superscript "p" standard notation? I think I usually see L_p as || ||_p (subscript only).
>
> As we noted in the text, $\| \cdot \|_p^p$ denotes the $L_p$-norm raised to the $p$-th power (e.g. $(|| ||_p)^p$, ).
>
> ---
> > __Re.11)__ Covariate matrix C: does this raise dimensionality issues, since C includes d^2 new parameters per filter? (doubles the number of free parameters I think).
>
> We would like to clarify that, $d$ indicates the spatial dimensionality, and in this paper $d=2$. Thus, $C$ is 2$\times$2 matrix. For the doubling of free parameters, we have tried our best to explain details in __Re.22)__.
>
> ---
> > __Re.12)__ Fig 1 e: this shows up too small on a printed page.
>
> Thank you. We increased the size of Fig. 1e in the revised manuscript.
>
> ___
> > __Re.13)__ Fig 1 f: Did you try p < 2? It looks like the masks converge to rectangles (all weights in mask the same) at low p, so that the difference between 4 and 16 is slight.
>
> Thank you for pointing out the behavior of masks at lower values of $p$. In our appendix (Appendix A.16), we have visualizations for $p<2$. Specifically, $p=1$, the mask forms a diamond shape, and for $p$ less than 1, it shifts to a cross-like shape. This change highlights the influence of  $p$ on the mask geometry. In fact, during the training, $p$ lower than 2 can also emerge.
>
> While the masks for $p=4$ and $p=16$ may appear similar, converging towards rectangles, as shown in Table 1, these values as initialization parameters can still lead to significant differences in performance. This suggests that even subtle variations in the shape of the mask can have a meaningful impact on the model's effectiveness.
>
> ---
> > __Re.14)__ Just after eqn 4 "C and p are trainable": I do not see how to do this. Is it explained in the paper?
>
> We used standard backpropagation, for detail see __Re. 2)__.
>
> ---
> > __Re.15)__ "mask take one": I stumbled on this. Clearer might be "equal one", or "have the value one". Also, maybe note that this is the limit as p increases, and in fact is roughly attained at p = (some value).
>
> We appreciate your review.
> First, we have revised our terminology to use “equal one” instead of “mask take one”.
> Second, theoretically, $L_{p}$ mask converges to a binary mask as p approaches infinity. However, as shown in Fig. 1, we observe that $L_{p}$ mask becomes a binary mask for sufficiently large $p$. We have included this discussion in the second last paragraph of Section 3.

---

> ### Author Response · Authors · 2023-11-20
> **Official Comment by Authors - 3**
>
> > __Re.16)__ Sudoku: Perhaps explain why this is a good test case for the method (since it is not usual, and the usual image examples come later).
>
> In response to your observation, we initially observed masks trained with usual images (as shown in __Figure 2 f and g__). This led us to conclude that the adaptability of $L_p$-masks could be advantageous due to their ability to create varied structured RFs. We identified the Sudoku task as an effective way to highlight these benefits, especially in our ablation study comparing column and row accuracy. Presenting the Sudoku task first was a strategic choice to help readers understand the $L_p$-convolution mechanism before demonstrating its performance enhancements and brain alignment.
>
> ---
> > __Re.17)__ Fig 3 b: the line colors/labels are unclear in these subplots. Perhaps make them bigger, or use dashed lines for some cases, or change to brighter colors.
>
> Thank you for the suggestion. We’ve changed with bigger, dashed lines.
>
> ---
> > __Re.18)__ Fig 4: What is the definition of "distortion"?
>
>
> In this work, the concept of “distortion”, denoted by $\gamma$, is introduced to quantify the extent of deformation in the shape of a mask. This parameter specifically measures the degree of asymmetry in the mask. A symmetric mask is characterized by $\gamma = 1$, and as $\gamma$ increases, the mask becomes increasingly asymmetric.
>
> Upon performing Singular Value Decomposition (SVD) on a $2 \times 2$ covariance matrix $\mathbf{C}$ of the distribution, two singular values, $\lambda_1$ and $\lambda_2$, are obtained. In a symmetric distribution, these singular values, $\lambda_1$ and $\lambda_2$, are identical. However, as the distribution becomes asymmetric, or distorted, these values diverge from each other (See __Appendix A.6__).
>
> ---
> > __Re.19)__ Fig 4 a: The different labels (alpha, theta) on the two sides of the mosaic are hard to decipher, especially on a printed page. Perhaps make the mosaic horizontal, or make labels larger, or break up the sets of 9 boxes with bolder lines, or provide clearer guidance in the caption.
>
> Thank you for the suggestion. We have enlarged Fig. 4a in the revised manuscript.
>
> ---
> > __Re.20)__ Fig 4 a: what role do diagonal masks play in the Sudoku networks, given the problem's horizontal-vertical-box structure?
>
> Thank you for this intriguing question. As observed in the 3rd column of Fig 4b, most diagonal masks have diminished, leading us to believe that diagonal masks do not play a significant role in the context of the Sudoku task. However, this observation presents an exciting opportunity for future experiments. For instance, in tasks like the BINGO game, which may require diagonal pattern recognition, diagonal masks could potentially play a crucial role.
>
> ---
> > __Re.21)__ Fig 4 a: A crucial question for me: These RFs look a lot like what we often see in CNN papers. Is the L_p method generating something novel, or is it a new way of getting to what already happens?
>
> $L_p$-Convolution enforces Rfs into explicit structural constraints, whereas RFs in conventional CNNs emerge naturally. Please see __Re. 1)__ for details.
>
> ---
> > __Re.22)__ Comparisons with base models: It appears that adding an L_p mask at each filter effectively doubles the free parameter count while holding the filter size fixed. Might this account for the difference in accuracy scores?
>
> Thank you for your insightful question. To clarify, "d" in this context refers to the dimension of the 2D space, which means the shape of C needed to create the mask is 2x2. To aid understanding, let's calculate the number of parameters for three cases, assuming there are "c" channels:
>
> 1. **(3x3), (Base) Conv**: The number of parameters here is c x 3 x 3, which equals 9c.
> 2. **(7x7), (Large) Conv**: For this model, the number of parameters is c x 7 x 7, totaling 49c.
> 3. **(7x7), $L_p$-Conv**: In this case, the number of parameters is c x (7 x 7 + 5) = 54c. The $L_p$-Convolution requires a mask for each channel, and each mask has 5 free parameters (comprising of "p" and the elements of the C matrix, which is 2x2), leading to an additional c x 5 parameters.
>
> In the above example, the free parameter increase in '(Base)Conv' to '(Large)Conv' is over __x5__ (9c vs. 49c) whereas '(Large)Conv' to '$L_p$-Conv' is __x1.1__. However, simply increasing free parameters doesn't guarantee the performance gain, as can be seen in row 1 vs row 2 in Table 1. However, $L_p$-Conv parameters stably achieve performance gains with a small increase in free parameters. This analysis suggests that factors beyond just parameter count are contributing to the observed differences in performance.
>
> ---
> > __Re.23)__ Section 7: This looks like it should appear earlier in the paper - it is background.
>
> In response to your comment, Section 7 was initially upfront but was later repositioned to improve the paper's readability and logical progression. Thank you for your recommendation. We deeply appreciate your thorough review!

---

> > ### Comment · Reviewer_Bf1L · 2023-11-22
> > **Response to Authors' responses**
> >
> > My thanks to the authors for their comprehensive responses to all the reviewers. They reinforce my positive assessment as to the quality of the paper.

---

### Official Review · Reviewer_UBiz · 2023-11-01

**Soundness:** 4 excellent
**Presentation:** 4 excellent
**Contribution:** 3 good
**Rating:** 8
**Confidence:** 3

**Summary:**

In this paper, large convolution kernels are masked by multiplication with a parameterizable function that can range from a 2D Gaussian to a more box-like function. The mask parameters are trained with the rest of the network. Suitable masks are learned in a network that solves Sudoku puzzles. Adding this method to several common architectures improves their image classification performance. Activity in these trained networks is compared with mouse visual cortex activity via representational similarity analysis, and it is found that more Gaussian-like masks tend to produce higher peak similarities between mouse and model representations.

**Strengths:**

The method is interesting, elegant, effective, and as far as I know novel. The paper is clear and well organized. The experiments make sense for demonstrating several different properties of the model, and the results seem convincing.

**Weaknesses:**

I don’t find much to complain about but I will do my best.

It’s interesting that the mask is more general than a Gaussian function, but the benefits of non-Gaussian versions seem less clear (e.g. Table 1). I don’t know whether I should actually use this mask rather than a simpler Gaussian one.

The p value that sets the smoothness of the mask seems not to change much during training, judging by the visualizations in Figure 2 and the performance differences due to different initializations of p in Table 1. I wasn’t sure that it was actually being optimized effectively. If it were optimized more successfully then it seems that a method other than initialization might be needed to apply pressure on p one way or another (such as a loss term).

**Questions:**

In the Figure 1g caption please clarify which comparisons were tested and whether there was a correction for multiple comparisons.

Page 8 says the models were compared with data from multiple subregions of mouse V1, but the appendix shows VISal, VISam, etc. as well as VISp. Is V1 a typo on page 8?

Please elaborate on the logic of the definition of functional synapses in the 3rd paragraph of A.2. Is it meant to relate to biological functional synapses? How?

In Figure 7 b and c, the middle-row (untrained) receptive fields have a grid structure and the ones in the bottom rows (trained) mostly have a bright line along the bottom. Could these phenomena please be explained?

---

> ### Author Response · Authors · 2023-11-20
>
> Thank you for your positive and constructive feedback! We've further improved the clarity and readability of our paper. Please find our responses to each suggestion below.
>
> ---
> > __Re.1)__ It’s interesting that the mask is more general than a Gaussian function, but the benefits of non-Gaussian versions seem less clear (e.g. Table 1). I don’t know whether I should actually use this mask rather than a simpler Gaussian one.
>
> We appreciate your comment. While we recommend using the Gaussian version (p=2) as shown in Table 1 for its overall effectiveness, note, that specific scenarios, such as with AlexNet, benefit more from a non-Gaussian version (p=16). This suggests that the Gaussian version isn't universally superior.
>
> ---
> > __Re.2)__ The p-value that sets the smoothness of the mask seems not to change much during training, judging by the visualizations in Fig. 2 and the performance differences due to different initializations of p in Table 1. I wasn’t sure that it was actually being optimized effectively. If it were optimized more successfully then it seems that a method other than initialization might be needed to apply pressure on p one way or another (such as a loss term).
>
> Appreciate your feedback. The result attached in __Appendix A.17__ and table below confirms that the parameter $p$ is actively optimized during training:
>
> | |Lp2|Lp4|Lp8|Lp16|
> |-|-|-|-|-|
> |CIFAR100|1.80~2.07|3.59~4.14|7.36~8.25|14.99~16.73|
> |TINYIMNET|1.72~2.08|3.26~4.10|6.61~8.47|13.77~18.86|
>
> However, the variation in post-training $p$ values was not dynamic enough to oscillate across different initialization conditions. Thus, your idea of applying a dedicated loss function to identify the optimal $p$ is very appealing. Thank you for the clever idea!
>
> ---
> > __Re.3)__ In the Fig.1g caption please clarify which comparisons were tested and whether there was a correction for multiple comparisons.
>
> We appreciate the reviewer's expert insight. Since we didn't correct for multiple comparisons, we conducted Holm-Bonferroni correction[1] in __Appendix A.19__.
>
> |p-values|Mouse V1|Train+Image|Train+Noise|Untrain+Noise|Untrain+Image|
> |-|-|-|-|-|-|
> |**Mouse V1**|-|*|*|***|***|
> |**Train+Image**|0.0110|-|n.s|***|***|
> |**Train+Noise**|0.0106|0.76|-|***|***|
> |**Untrain+Noise**|1.23e-15|4.13e-28|1.49e-25|-|***|
> |**Untrain+Image**|8.94e-14|1.59e-25|6.73e-23|1.39e-22|-|
>
> We believe our results are now more robust and solid with the suggested correction.
> We also updated the Fig. 1g caption as follows:
>
> “Using Welch's t-test with Holm-Bonferroni's multiple comparisons correction, all possible combinations between groups were statistically significant (p-value$<$0.05) except for `n.s.' (non-significant) denoted in the figure.”
>
> [1] Holm, Sture. "A simple sequentially rejective multiple test procedure." Scandinavian journal of statistics (1979).
>
> ---
> > __Re.4)__ Page 8 says the models were compared with data from multiple subregions of mouse V1, but the appendix shows VISal, VISam, etc. as well as VISp. Is V1 a typo on page 8?
>
> Thank you for the correction. The correct term is 'VC', not 'V1', and we revised notations including  __Section 6, Appendix A.12  and Fig. 5__. We are grateful for your expertise and for guiding us to address our areas of deficiency.
>
> ---
> > __Re.5)__ Please elaborate on the logic of the definition of functional synapses in the 3rd paragraph of A.2. Is it meant to relate to biological functional synapses? How?
>
> Yes, the definition aligns with biological functional synapses, facilitating a direct comparison between biological and artificial systems. In biological contexts, functional synapses are often identified through Z-score-based activity thresholding in calcium recordings, as noted in our revised manuscript's footnote 5 under section A.2. We adopted this approach for our artificial system.
>
> To clarify this concept, we
> - added new figures illustrating functional synapses in __Appendix A.14 &A.15__,
> - clarified the wording in the third paragraph of __Appendix A.2__, as follows: "Since spatial connectivity pattern in the biological synapse is measured by the functional calcium activities and given as coordinates, we applied a similar approach to that of CNN layers."
>
> ---
> > __Re.6)__ In Fig. 7 b&c, the middle-row (untrained) receptive fields have a grid structure and the ones in the bottom rows (trained) mostly have a bright line along the bottom. Could these phenomena please be explained?
>
> Thank you for pointing out these phenomena. In the trained model, the bright line could be a result of typical shading patterns in images, as illustrated in Fig. 7b. Its presence in noise inputs, however, hints at potential training limitations related to lack of rotation or vertical flip augmentations. For the untrained model, the grid pattern, as shown in Appendix A.15, Fig. 12d, is an ongoing area of study. Your question has importantly highlighted the role of top-down shading in image processing, paving the way for further exploration.

---

> > ### Comment · Reviewer_UBiz · 2023-11-22
> >
> > Thank you for your responses. They address my questions well, except that the last point still seems unclear. Overall I think the paper has improved mainly in response to other reviewers' comments (\which I appreciated).

---

### Official Review · Reviewer_YKuY · 2023-11-02

**Soundness:** 4 excellent
**Presentation:** 4 excellent
**Contribution:** 4 excellent
**Rating:** 8
**Confidence:** 3

**Summary:**

The Authors introduce the so-called Lp-convolution that is based on the multivariate p-generalized normal distribution to address the gap between the artificial and biological receptive fields. The Authors study the properties of the Lp-convolution and provide evidence that the proposal benefits, for instance, large kernel sizes in a classification task using previous well-studied architectures (e.g., AlexNet) with the CIFAR-100 and TinyImageNet datasets.

**Strengths:**

The manuscript has been well-written, and the ideas behind the paper have been legibly presented with an experimental part following the requirements of the ICLR conference.

I have found the most attractive part of the paper in section 6, where The Authors evaluated the alignment between biological
and artificial models.

**Weaknesses:**

I do not see particular weaknesses in the manuscript.

**Questions:**

Could the Authors comment on how their work is related to the concept of foveation?

https://doi.org/10.1007/s11263-016-0898-1

---

> ### Author Response · Authors · 2023-11-20
>
> Thank you to the reviewer for the highly encouraging feedback and for finding merit in our brain alignment experiment!
> We also express our gratitude for the introduction of the following insightful paper.
>
> ---
> > __Re.1)__ Could the Authors comment on how their work is related to the concept of foveation? https://doi.org/10.1007/s11263-016-0898-1
>
> We thank the reviewer for introducing this insightful paper. It has been enlightening to read and has significantly influenced our future research directions, encouraging us to explore new methodologies. This paper and our work share a common theme of adopting concepts from human visual processing to enhance computational techniques.
>
> Operationally, both approaches modify the visual receptive area (patch or kernel) as it moves away from the center fixation point. __While Lp-convolution (p=2) can be seen as reducing brightness, Foveation applies blurring.__
>
> These differences, stemming from __the initial stages of human visual processing in Foveation__ and __the local connectivity patterns of the visual cortex in Lp-convolution__, raise an exciting question: ___Could a blend of these approaches further enhance visual processing of artificial models?___ We are excited to explore this possibility in the future direction of our research.

---

### Official Review · Reviewer_oxDe · 2023-11-02

**Soundness:** 3 good
**Presentation:** 1 poor
**Contribution:** 2 fair
**Rating:** 5
**Confidence:** 3

**Summary:**

In this submission, the authors propose to bridge the gap between artificial and biological visual receptive fields by introducing $L_p$ convolutions implemented modeled using the multivariate p-generalized normal distribution (MPND). The authors claim that it is possible to model a spectrum of receptive fields with increasing resemblance to biological receptive fields by tuning the $p$ and $\sigma$ parameters of MPND. On a Sudoku quiz benchmark, the authors show that L-p convolution is capable of learning diversely shaped receptive fields. On a couple of image classification benchmarks (CIFAR-100 and Tiny ImageNet), the authors show that tuning $p$ in various convolutional architectures integrated with $L_p$ convolution leads to classification performance gains. Representational similarity analysis testing the neural encoding ability of $L_p$ convolutional networks at different values of $p$ shows that networks with smaller $p$ (which are characteristic of biological RFs modeled with $L_p$ convolution) are better encoders of mouse visual representations.

**Strengths:**

+ This submission proposes a very interesting characterization of the disparities between artificial and biological receptive fields using multivariate p-generalized normal distributions (MPNDs).
+ The authors have worked rigorously on their model comparison experiments by testing all models with different random initializations and adequate statistical testing to highlight significant differences.
+ It is really interesting that $L_p$ convolutions with smaller $p$ values are also better encoders of mouse visual representations recorded from V1 in response to natural images.
+ Thanks to the authors for releasing code for reproducing their results.

**Weaknesses:**

- The paper is quite hard to read. Especially the sections introducing $L_p$ convolution (sections 2 and 3) need to be written with much more clarity to make them more accessible. Currently, there are issues such as an abundance of notations, symbols being introduced after their first use, etc. that make it difficult to understand exactly how $L_p$ convolution works and models the spectrum from biologically resembling to artificial receptive fields.
- I don't find the similarity of $L_p$ convolution with small p with mouse visual receptive fields to be very convincing. First of all, this seems like a qualitative comparison and is not an objective way to measure similarity to biological receptive fields. It also seems from Appendix A4 (Fig 7A) that the mouse functional synapses in V1 lack any visible structure representative of selectivity to low-level visual features. This is quite concerning as it makes one wonder whether these neurons are selective to low-level features as one would expect from V1 neurons.
- There also seems to be an issue with both untrained and trained receptive fields of AlexNet's conv1 layer in the same figure. Untrained filters seem to have a peculiar checkerboard-like structure that one wouldn't expect in randomly-initialized kernels. Re. pretrained filters, in the AlexNet paper [1] the authors plot the filters in the first convolution layer of an AlexNet trained on ImageNet-1k in Figure 3 of their paper. There is a big gap in terms of how selective their filters are in comparison to the filters visualized in the current submission in Figure 7 of Appendix A4. Could the authors please explain this discrepancy?
- Overall, I believe that in the current state, there are several open issues such as the (key) ones I highlighted here in my review that need to be fixed in order to push this paper above the acceptance threshold.

References:
1. Krizhevsky, A., Sutskever, I., & Hinton, G. E. (2012). Imagenet classification with deep convolutional neural networks. Advances in neural information processing systems, 25.

**Questions:**

Please refer to the weaknesses section in my review above.

---

> ### Author Response · Authors · 2023-11-20
>
> Thank you for your valuable feedback and guidance. We have significantly enhanced the clarity and readability of our work in response to the reviewers' comments. Below are our point-by-point responses to each suggestion.
>
> ---
> > __Re.1)__ The paper is quite hard to read. Especially the sections introducing $L_p$ convolution (sections 2 and 3) need to be written with much more clarity to make them more accessible. Currently, there are issues such as an abundance of notations, symbols being introduced after their first use, etc. that make it difficult to understand exactly how $L_p$ convolution works and models the spectrum from biologically resembling to artificial receptive fields (RFs).
>
> We appreciate your review. We have revised Section 2 and Section 3.
> First, we have simplified the content by excluding notations that were previously unused, and have added clear explanations for all notations that are used. (e.g., we remove $\varphi$, and we add description for $C$ and $W_i$)
> Second, we clarify the relationship between RF and $L_p$-convolution in the paragraph below Equation 3.
>
> ---
> > __Re.2)__ I don't find the similarity of $L_p$ convolution with small p with mouse visual RFs to be very convincing. First of all, this seems like a qualitative comparison and is not an objective way to measure similarity to biological RFs. It also seems from Appendix A4 (Fig 7A) that the mouse functional synapses in V1 lack any visible structure representative of selectivity to low-level visual features. This is quite concerning as it makes one wonder whether these neurons are selective to low-level features as one would expect from V1 neurons.
>
> Thank you for your insightful feedback regarding the comparison of $L_p$-convolution with small $p$ to mouse visual RFs. We appreciate your concern about the qualitative nature of this comparison and the observations from Appendix A4 (Fig 7A).
> In response to your comments, we would like to clarify our approach and make some adjustments to our manuscript for better clarity. ___We realize that our definition of "receptive fields" might have been misleading.___ Our definition aligns more with the engineering perspective prevalent in CNNs, especially for AI/ML audience at ICLR. Our adjustments in the manuscript are as follows:
>
> __Clarify the definition in the main text (Section 2):__ To clarify our definition of RF, we promoted footnote 1 to the main text, with bold highlighted at local connectivity patterns.
> - (new): While the concept of RF generally encompasses sensory-level inputs and adjacent layers, in this paper, we specifically refer to __RF as local connectivity patterns__ between neurons in adjacent layers.
>
> __Clarify the description in the appendix (Figure 7 legend):__ To further clarify, we have inserted a note in the figure legend.
> - (new): The receptive field discussed in this figure specifically refers to the spatial connectivity patterns of synapses. Note that this differs from receptive fields typically associated with low-level visual feature selectivity.
>
> ---
> > __Re.3)__ There also seems to be an issue with both untrained and trained RFs of AlexNet's conv1 layer in the same figure. Untrained filters seem to have a peculiar checkerboard-like structure that one wouldn't expect in randomly-initialized kernels. Re. pretrained filters, in the AlexNet paper [1] the authors plot the filters in the first convolution layer of an AlexNet trained on ImageNet-1k in Fig. 3 of their paper. There is a big gap in terms of how selective their filters are in comparison to the filters visualized in the current submission in Fig. 7 of Appendix A4. Could the authors please explain this discrepancy?
>
> We appreciate the thorough comments from the reviewer, grounded in deep insights. The visualization of kernels in Fig. 3 in the original AlexNet paper and our functional synapse connectivity pattern as RF differ due to methodological distinctions.
>
> To clearly convey the main difference, we have added additional figures in revised manuscript __Appendix A.14 & A.15__.
> - Observing __Fig. 11 a&b__, one can see kernel RFs similar to those in the AlexNet paper Fig. 3. In this study, the functional RF we refer to is an effort to visualize and observe the functionally active spatial connectivity pattern when actual input is present.
> - __Fig. 11 d&e, and Fig. 12 d&e__ correspond to individual images in __Appendix A.4__. We hope our explanation sufficiently resolves the discrepancy between the original AlexNet RF and ours.
>
> ---
> > __Re.4)__ Overall, I believe that in the current state, there are several open issues such as the (key) ones I highlighted here in my review that need to be fixed in order to push this paper above the acceptance threshold.
>
> We believe the key issue arises from our definition of RFs in the context of CNNs from an engineering perspective. We hope that these revisions and clarifications will resolve any misunderstandings and align our findings more closely with the audience.

---

> ### Author Response · Authors · 2023-11-22
> **A gentle reminder**
>
> Dear reviewer,
>
> We've updated our manuscript in response to your valuable feedback, focusing on improving the accessibility of sections 2 and 3 and clarifying our definition of the receptive field. We're keen to know if these revisions align with your expectations. Your further thoughts would be greatly appreciated.

---

> ### Comment · Reviewer_oxDe · 2023-11-30
> **Thank you for your response to our reviews**
>
> Dear authors,
>
> Thank you for your response to our reviews. The additional clarification is helpful, however, my major concerns about the work still persist. I would like to differ from the authors opinion that the key issue is in the definition of RFs, I am hence retaining my not-accept rating.
>
> However, as other reviewers have pointed out that the paper is well-written, I stand corrected and this is potentially an issue that I particularly had due to less familiarity with the specific methods in the paper. As a result, I raised my rating from 3 -> 5, although I still think the paper could be further improved before being accepted at ICLR.
>
> Thank you.

---

### Author Response · Authors · 2023-11-20

__Summarized Strengths__

We are grateful for the reviewers' encouraging comments, emphasizing several aspects of our study. The unanimous agreement on the innovative and interesting nature of our idea stands out (__all Reviewers__). The paper's clarity and structure were well-received (__Reviewers YKuY, UBiz, Bf1L, qr9c__), and the thoroughness of our experimental work was convincingly recognized (__Reviewer oxDe, YKuY, UBiz, qr9c__), with particular praise for our statistical validation approach (__Reviewer oxDe & Bf1L__). The brain alignment results were also highlighted as a noteworthy feature (__Reviewer oxDe, YKuY, Bf1L__), and the release of our code was appreciated for ensuring reproducibility (__Reviewer oxDe & Bf1L__).

__Summarized Feedback__

The reviewers have provided critical and insightful feedback, which has been instrumental in the direction of refining our study.
In summary, reviewers
- suggested a clearer explanation and justification of the "Receptive Field" concept in our study (__Reviewer oxDe & qr9c__ ).
- recommended a more audience-friendly presentation of our methodologies, particularly in Sections 2 & 3 (__Reviewer oxDe__).
- advised providing more comprehensive details on the characteristics of the trained $L_p$-mask (__Reviewer UBiz & qr9c__).
- emphasized the need for stronger statistical validation, including adjustments for multiple comparisons (__Reviewer UBiz__).
- asked further clarification of advantages on $L_p$-Convolution (__Reviewer qr9c__).

__Summarized Changes__

Thanks to the insightful feedback provided by reviewers, our revised manuscript has been greatly improved.
The major changes made in our revised manuscript are
- clarifying the definition, and description in the main and appendix (__Blue colors in text__).
- presenting new figures that illustrate the functional connectivity patterns of CNNs (__Fig. 11 & 12 of Appendix A.14 & 15__).
- showcasing visualizations of MPND structure for $p$ values under 2 (__Fig. 13 of Appendix A.16__).
- detailing the distributions of $p$ in $L_p$-masks after training (__Figs. 14 & 15, Tables 2 & 3 of Appendix A.17__).
- showing a layer-wise properties of the $L_p$-mask (p=16) in Sudoku training (__Fig. 16 of Appendix A.18__).
- correcting the statistical test result with Horm-Bonferroni method (__Table 4 of Appendix A.19__).
- additional experiments with $L_p$-Convolution other CNN architectures  (__Table 5 of Appendix A.20__).

We are deeply thankful to the reviewers for their insightful feedback and dedicated efforts. Thanks to their contributions, our revised paper now aligns more closely with the high standards of ICLR.  ___Please find our revised paper for the above results and more details___.

---

### Meta-Review · Area_Chair_JM8f · 2023-12-11

**Metareview:**

In this work, the authors propose a new form of convolutional neural networks which employs a form of a convolution that is more similar to biological visual receptive fields. In particular, the authors propose a Gaussian masking strategy for convolution filters for image classification tasks which leads to the ability to learn larger receptive field filters. The authors find that the learned filters more closely match the representation learned in the mouse visual system and improves a CNN performance on a Sudoku challenge as well as several small scale image classification tasks (e.g. CIFAR, Tiny-ImageNet).

The reviewers commented positively on the creative inductive bias the authors employed, the quality of the presentation of the work and the well-motivated and well-executed experiments. The reviewers commented negatively on the marginal gains in terms of representational similarity and performance on the Sudoku and image classification tasks. Additionally, the reviewers surfaced some concerns about the experimental details.

The authors provided several responses to address these concerns including pointing to new results on several additional architectures. One reviewer upgraded their score but two reviewers still felt the paper was below acceptance threshold given these concerns. In particular, the reviewer still felt unconvinced that the authors have successfully demonstrated a strong similarity with the mouse visual receptive fields. Additionally, the gains while statistically significant appear quite small and it is unclear what this means in terms of the utility of the method let alone the correspondence to the underlying biology.

Granted, there was a split viewpoint between the reviewers. Several reviewers argued for acceptance while others argued for rejection. Given the lack of consensus, I took a closer look at the paper. In my reading of the paper, I found that the reviewers arguing for lower scores presented notable concerns in terms of the marginal gains of the method and uncertainty as to whether or not the method demonstrated any strong correspondence with the underlying biology. I was not able to find reason in the other reviewers comments which outweighed these concerns. Although I applaud the author’s efforts in their motivation in bridging biology and machine learning, I must agree with the more critical reviewers and will not accept this paper to this conference.

**Justification For Why Not Higher Score:**

Marginal gains. Unclear what this says about biology.

**Justification For Why Not Lower Score:**

N/A

---

### Decision · Program_Chairs · 2024-01-16

Reject